# Hierarchical and Density-based Causal Clustering

**Kwangho Kim**
Korea University
kwanghk@korea.ac.kr

**Jisu Kim**
Seoul National University
jkim82133@snu.ac.kr

**Larry A. Wasserman**
Carnegie Mellon University
larry@stat.cmu.edu

**Edward H. Kennedy**
Carnegie Mellon University
edward@stat.cmu.edu

## Abstract

Understanding treatment effect heterogeneity is vital for scientific and policy research. However, identifying and evaluating heterogeneous treatment effects pose significant challenges due to the typically unknown subgroup structure. Recently, a novel approach, causal k-means clustering, has emerged to assess heterogeneity of treatment effect by applying the k-means algorithm to unknown counterfactual regression functions. In this paper, we expand upon this framework by integrating hierarchical and density-based clustering algorithms. We propose plug-in estimators which are simple and readily implementable using off-the-shelf algorithms. Unlike k-means clustering, which requires the margin condition, our proposed estimators do not rely on strong structural assumptions on the outcome process. We go on to study their rate of convergence, and show that under the minimal regularity conditions, the additional cost of causal clustering is essentially the estimation error of the outcome regression functions. Our findings significantly extend the capabilities of the causal clustering framework, thereby contributing to the progression of methodologies for identifying homogeneous subgroups in treatment response, consequently facilitating more nuanced and targeted interventions. The proposed methods also open up new avenues for clustering with generic pseudo-outcomes. We explore finite sample properties via simulation, and illustrate the proposed methods in voting and employment projection datasets.

## 1 Introduction

### 1.1 Heterogeneity of Treatment Effects

Causal effects are typically summarized using population-level measures, such as the average treatment effect (ATE). However, these summaries may be insufficient when treatment effects vary across subgroups. For example, finding the subgroups that experience the least or greatest benefit from a specific treatment is of particular importance in personalized medicine or policy evaluation, where the subgroup effects of interest may diverge significantly from the population effect. Even while experiencing the same treatment effects, some people may have been exposed to a significantly higher baseline risk. In the presence of effect heterogeneity, the typically unknown subgroup structure poses significant challenges in accurately identifying and evaluating subgroup effects compared to population-level effects.

To delve deeper than the information provided by the population summaries and to better understand treatment effect heterogeneity, investigators often estimate the conditional average treatment effect (CATE) defined by

$$\mathbb{E}(Y^1 - Y^0 \mid X),$$

where $Y^a$ is the potential outcome that would have been observed, possibly contrary to fact, under treatment $A = a$, and $X \in \mathcal{X}$ is a vector of observed covariates. The estimation of the CATE has the potential to facilitate the personalization of treatment assignments, taking into account the characteristics of each individual. Admittedly, the CATE is the most commonly-used estimand to study treatment effect heterogeneity. Various methods have been proposed to obtain accurate estimates of and valid inferences for the CATE, with a special emphasis in recent years on leveraging the rapid development of machine learning methods [e.g., 3, 20, 21, 27, 35, 42, 46, 52, 53, 58, 62].

Subgroup analysis has been the most common analytic approach for examining heterogeneity of treatment effect. Selection of subgroups reflecting one's scientific interest plays a central role in the subgroup analysis. Statistical methods aimed at finding such subgroups from observed data have been termed *subgroup discovery* [43]. The selection of such subgroups may be informed by mechanisms and plausibility (e.g., clinical judgment), taking into account prior knowledge of treatment effect modifiers. They could be chosen by directly subsetting the covariate space, often in a one-variable-at-a-time fashion [e.g., 49]. Most existing studies on data-driven subgroup discovery identify subgroups where the CATE exceeds a prespecified threshold of clinical relevance, allowing researchers to prioritize subgroups with enhanced efficacy or favorable safety profiles [e.g., 6, 11, 44, 47, 51, 63]. Some recent advances proposed heuristics for discovering rules based on a specific CATE estimator subject to a certain optimality criterion, yet without any theoretical exploration [e.g., 8, 15, 23, 48]. Wang and Rudin [59] proposed an algorithm to automatically find a subgroup based on the causal rule: (CATE > ATE). Kallus [31] proposed a subgroup partition algorithm for determining a subgroup structure that minimizes the personalization risk.

## 1.2 Causal Clustering

In contrast to earlier work predominantly focused on supervised learning approaches, there is a growing interest in analyzing heterogeneity in causal effects from an unsupervised learning perspective, particularly within the causal discovery literature. Based on the causal graph or structural causal model framework, there has been a series of recent attempts to learn *structural heterogeneity* through clustering analysis [e.g., 25, 26, 41, 45]. Conversely, the exploration of *treatment effect heterogeneity* in the potential outcome/counterfactual framework using unsupervised learning methods has received significantly less attention. To our knowledge, only one paper has developed such methods; Kim et al. [39] have proposed *Causal k-Means Clustering*, a new framework for exploring heterogeneous treatment effects leveraging tools from cluster analysis, specifically k-means clustering. It allows one to understand the structure of effect heterogeneity by identifying underlying subgroups as clusters without imposing a priori assumptions about the subgroup structure.

To illustrate, we consider binary treatments and project a sample onto the two-dimensional Euclidean space $(\mathbb{E}[Y^0 \mid X], \mathbb{E}[Y^1 \mid X])$. It is immediate to see that closer units are more homogeneous in terms of the CATE, which provides vital motivation for uncovering subgroup structure via cluster analysis on this particular counterfactual space. (See (a) & (e) in Figure 1). This approach has the capability to uncover complex subgroup structures beyond those identified by CATE summary statistics or histograms. Moreover, it holds particular promise in outcome-wide studies featuring multiple treatment levels [56, 57], because instead of probing a high-dimensional CATE surface to assess the subgroup structure, one may attempt to uncover lower-dimensional clusters with similar responses to a given treatment set.

However, the method proposed by Kim et al. [39] only applies to k-means clustering. Despite is popularity, k-means has some drawbacks. It works best when clusters are at least roughly spherical. It also has trouble clustering data when the clusters are of varying sizes and density, or based on non-Euclidean distance. Furthermore, the cluster centers (centroids) might be dragged by outliers, or outliers might even get their own cluster. Other commonly-employed clustering algorithms, particularly hierarchical and density-based approaches, could mitigate some of these limitations [1, 9, 22, 29, 40]. Density-based clustering is applicable for identifying clusters of arbitrary sizes and shapes, while concurrently exhibiting robustness to noise and outlier data points. Hierarchical clustering proves beneficial in scenarios where the data exhibit a nested structure or inherent hierarchy, irrespective of their shape, and can accommodate various distance metrics. It enables for the creation of a dendrogram, which provides insights into the interrelations among clusters across multiple levels of granularity. Figure 1 illustrates the three methods in the causal clustering framework with

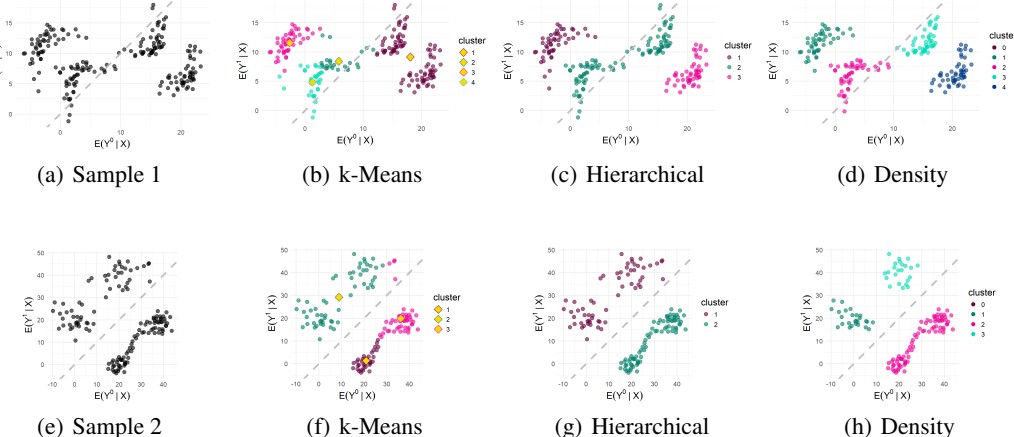

Figure 1: Two instances in which the three clustering techniques result in distinct subgroups for the projected sample. The grey dotted diagonal line indicates no treatment effects.

binary treatments, where hierarchical and density-based clustering methods produce more reasonable subgroup patterns.

In this work, we extend the work of Kim et al. [39] by integrating hierarchical and density-based clustering algorithms into the causal clustering framework. We present plug-in estimators which are simple and readily implementable using off-the-shelf algorithms. Unlike k-means clustering, which requires the margin condition [39], our proposed estimators do not rely on such strong structural assumptions on the outcome process. We study their rate of convergence, and show that under the minimal regularity conditions, the additional cost of causal clustering is essentially the estimation error of the outcome regression functions. Our findings significantly extend the capabilities of the causal clustering framework, thereby contributing to the progression of methodologies for identifying homogeneous subgroups in treatment response, consequently facilitating more nuanced and targeted interventions. In a broader sense, causal clustering may be construed as a nonparametric approach to clustering involving unknown functions, a domain that has received far less attention than conventional clustering techniques applied to fully observed data, notwithstanding its substantive importance. Therefore, the proposed methods also open up new avenues for clustering with generic pseudo-outcomes that have never been observed, or have been observed only partially.

## 2   Framework

Following Kim et al. [39], we consider a random sample $(Z_1, ..., Z_n)$ of $n$ tuples $Z = (Y, A, X) \sim \mathbb{P}$, where $Y \in \mathbb{R}$ represents the outcome, $A \in \mathcal{A} = \{1, ..., q\}$ denotes an intervention with finite support, and $X \in \mathcal{X} \subseteq \mathbb{R}^d$ comprises observed covariates. For simplicity, we focus on univariate outcomes, although our methodology can be easily extended to multivariate outcomes. Throughout, we rely on the following widely-used identification assumptions:

**Assumption C1** (consistency). $Y = Y^a$ if $A = a$.

**Assumption C2** (no unmeasured confounding). $A \perp\!\!\!\perp Y^a \mid X$.

**Assumption C3** (positivity). $\mathbb{P}(A = a \mid X)$ is bounded away from 0 a.s. $[\mathbb{P}]$.

For $a \in \mathcal{A}$, let the outcome regression function be denoted by

$$\mu_a(X) = \mathbb{E}(Y^a \mid X) = \mathbb{E}(Y \mid X, A = a).$$

Then, the pairwise CATE can be consequently defined as $\tau_{aa'}(X) = \mu_a(X) - \mu_{a'}(X)$ for any pair $a, a' \in \mathcal{A}$. The *conditional counterfactual mean vector* $\mu : \mathcal{X} \to \mathbb{R}^q$ projects a unit characteristic onto a $q$-dimensional Euclidean space spanned by the outcome regression functions $\{\mu_a\}$:

$$\mu(X) = [\mu_1(X), \ldots, \mu_q(X)]^\top . \tag{1}$$

Adjacent units in the above counterfactual mean vector space would have similar responses to a given set of treatments by construction. If all coordinates of a point $\mu(X)$ are identical for a given $X$, it indicates the absence of treatment effects on the conditional mean scale. Hence, conducting cluster analysis on the transformed space by $\mu$ allows for the discovery of subgroups characterized by a high level of within-cluster homogeneity in terms of treatment effects. Crucially, standard clustering theory is not immediately applicable here since the variable to be clustered is $\mu$, a set of the unknown regression functions that must be estimated. We let $\{\widehat{\mu}_a\}$ be some estimators of $\{\mu_a\}$. In Sections 3 and 4, we analyze the nonparametric *plug-in* estimators for hierarchical and density-based causal clustering, respectively, where we estimate each $\mu_a$ with flexible nonparametric methods and perform clustering based on $\widehat{\mu} = (\widehat{\mu}_1, \ldots, \widehat{\mu}_q)^\top$.

It is worth noting that $\mu$ can be easily customized for a specific need through reparametrization, without affecting our subsequent results. For example, it is possible that the difference in regression functions may be more structured and simple than the individual components [e.g., 13, 35]. In this case, a parametrization such as $\mu = (\mu_2 - \mu_1, \mu_3 - \mu_1, \cdots)$ could render our clustering task easier by allowing us to harness this nontrivial structure (e.g., smoothness or sparsity) [see 39, Section 2].

**Notation.** We use the shorthand $\mu_{(i)} = \mu(X_i)$ and $\widehat{\mu}_{(i)} = \widehat{\mu}(X_i) = [\widehat{\mu}_1(X_i), ..., \widehat{\mu}_q(X_i)]^\top$. We let $\|x\|_p$ denote $L_p$ norm for any fixed vector $x$. For a given function $f$ and $r \in \mathbb{N}$, we use the notation $\|f\|_{\mathbb{P},r} = [\mathbb{P}(|f|^r)]^{1/r} = \left[\int |f(z)|^r d\mathbb{P}(z)\right]^{1/r}$ as the $L_r(\mathbb{P})$-norm of $f$. We use the shorthand $a_n \lesssim b_n$ to denote $a_n \leq \mathsf{c}b_n$ for some universal constant $\mathsf{c} > 0$. Further, for $x \in \mathbb{R}^q$ and any real number $r > 0$, we let $\mathbb{B}(x, r)$ denote an open ball centered at $x$ with radius $r$ with respect to $L_2$ norm, i.e., $\mathbb{B}(x, r) = \{y \in \mathbb{R}^q : \|x - y\|_2 < r\}$ and use the notation $\overline{\mathbb{B}(x, r)}$ for the closed ball. Lastly, we use the symbol $\equiv$ to denote equivalence relation between two notationally distinct quantities, especially when introducing a simplified notation.

## 3 Hierarchical Causal Clustering

Hierarchical clustering methods build a set of nested clusters at different resolutions, typically represented by a binary tree or dendrogram. Consequently, they do not necessitate a predetermined number of clusters and allow for the simultaneous exploration of data across multiple granularity levels based on the user's preferred similarity measure. Moreover, hierarchical clustering can be performed even when the data is only accessible via a pairwise similarity function. There are two types of hierarchical clustering: agglomerative and divisive. The agglomerative approach forms a dendrogram from the bottom up, finding similarities between data points and iteratively merging clusters until the entire dataset is unified into a single cluster. The divisive approach employs a top-down strategy, whereby clusters are recursively partitioned until individual data points are reached. Here we only consider the agglomerative approach which is more common in practice [61]. We remark that the similar argument in this section may be applicable to the divisive approach as well.

Consider a distance or dissimilarity between points, i.e., $d : \mathbb{R}^q \times \mathbb{R}^q \to [0, 1]$. As in previous studies [e.g., 14, 16, 30], we extend $d$ so that we can compute the distance, or *linkage*, between sets of points $S_1(\mu) \equiv S_1$ and $S_2(\mu) \equiv S_2$ in the conditional counterfactual mean vector space as $D(S_1, S_2)$. There are three common distances between sets of points used in hierarchical clustering: letting $N_1$ be the number of points in $S_1$ and similarly for $N_2$, we define the *single*, *average*, and *complete* linkages by $\min_{s_1 \in S_1, s_2 \in S_2} d(s_1, s_2)$, $\frac{1}{N_1 N_2} \sum_{s_1 \in S_1, s_2 \in S_2} d(s_1, s_2)$, and $\max_{s_1 \in S_1, s_2 \in S_2} d(s_1, s_2)$, respectively. Single linkage often produces thin clusters while complete linkage is better at spherical clusters. Average linkage is in between. Causal clustering entails estimating the nuisance regression functions $\{\mu_a\}$, which necessitates the following assumption.

**Assumption A1.** *Assume that either (i) $\{\mu_a\}$ and $\{\widehat{\mu}_a\}$ are contained in a Donsker class, or (ii) $\{\widehat{\mu}_a\}$ is constructed from a separate independent sample of same size.*

Assumption A1 is required essentially because in our estimation procedure, we use the sample twice, once for estimating the nuisance functions $\{\mu_a\}$ and again for determining the clusters. One may use the full sample if we restrict the flexibility and complexity of each $\widehat{\mu}_a$ through the empirical process conditions, as in (i), which may not be satisfied by many modern machine learning tools. In order to accommodate this added complexity from employing flexible machine learning, we can instead use sample splitting [e.g., 12, 64], as in (ii). We refer the readers to Kennedy [32, 33, 34] for more details.

In the following proposition, we give an error bound of computing the set distance with the conditional counterfactual mean vector estimates.

**Proposition 3.1.** *Let $D$ denote the single, average, or complete linkage between sets of points, induced by the distance function such that $d(x, y) \lesssim \|x - y\|_1$. Then under Assumption A1, for any two sets $S_1, S_2$ in $\{\mu_{(i)}\}$ and their estimates $\widehat{S}_1, \widehat{S}_2$ with $\{\widehat{\mu}_{(i)}\}$,*

$$\left| D(S_1, S_2) - D(\widehat{S}_1, \widehat{S}_2) \right| \lesssim \sum_{a \in \mathcal{A}} \|\widehat{\mu}_a - \mu_a\|_\infty.$$

A proof of the above proposition and all subsequent proofs can be found in Appendix. Proposition 3.1 suggests that in the agglomerative clustering we shall obtain identical cluster sets beyond a certain level of the dendrogram, where the distance between the closest pair of branches exceeds the outcome regression error. The result applies to a wide range of distance functions in Euclidean space.

In some problems it might be expensive to compute similarities between all $n$ items to be clustered (i.e., $O(n^2)$ complexity). Eriksson et al. [16] proposed the hierarchical clustering of $n$ items only based on an adaptively selected small subset of pairwise similarities on the order of $O(n \log n)$. By virtue of Proposition 3.1 their algorithm is also applicable to our framework as long as $\widehat{\mu}_a$ is a consistent estimator for $\mu_a$ [see 16, Theorem 4.1].

In contrast to k-means clustering, it is not straightforward to analyze the performance of hierarchical clustering with respect to the true target clustering, because we build a set of nested clusters across various resolutions (a hierarchy) such that the target clustering is close to some pruning of that hierarchy. Moreover, conventional linkage-based algorithms may have difficulties in the presence of noise. Balcan et al. [5] proposed a novel robust hierarchical clustering algorithm capable of managing these issues. Their algorithm produces a set of clusters that closely approximates the target clustering with a specified error rate even in the presence of noise, and it is adaptable to an inductive setting, where only a small subset of the entire sample is utilized. We shall adapt their algorithm for causal clustering, and analyze the performance of our proposed algorithm.

We consider an inductive setting where we only have access to a small subset of points from a much larger data set. This can be particularly important when running an algorithm over the entire dataset is computationally infeasible. Suppose that $\{C_1, ..., C_k\}$ is the target clustering, and that there exist $N$ samples in total. Assuming we are given a random subset $\mathsf{U}^n$ of size $n$, $n \ll N$, consider a clustering problem $(\mathsf{U}^n, l)$ in the conditional counterfactual mean vector space where each point $\mu \in \mathsf{U}^n$ has a true cluster label $l(\mu) \in \{C_1, ..., C_k\}$. Further we let $C(\mu)$ denote a cluster corresponding to the label $l(\mu)$, and $n_{C(\mu)}$ denote the size of the cluster $C(\mu)$. To proceed, we define the following *good-neighborhood* property to quantify the level of noisiness in our population distribution.

**Definition 3.1** (($\alpha, \nu$)-Good Neighborhood Property for Distribution). *For a fixed $\mu' \in \mathbb{R}^q$, let $\mathbb{C}(\mu') = \{\mu : C(\mu) = C(\mu')\}$, i.e., a set whose label is equal to $C(\mu')$, and $r_{\mu'} = \inf_r \{r : \mathbb{P}[\mu \in \mathbb{B}(\mu', r)] \equiv \mathbb{P}[\mathbb{C}(\mu')]\}$. The distribution $\mathbb{P}_{\alpha,\nu}$ satisfies $(\alpha, \nu)$-good neighborhood property if $\mathbb{P}_{\alpha,\nu} = (1 - \nu)\mathbb{P}_\alpha + \nu\mathbb{P}_{noise}$ where $\mathbb{P}_\alpha$ is a probability distribution that satisfies*

$$\mathbb{P}_\alpha\{\mu \in \mathbb{B}(\mu', r_{\mu'}) \setminus \mathbb{C}(\mu')\} \leq \alpha,$$

*and $\mathbb{P}_{noise}$ is a valid probability distribution.*

The good-neighborhood property in Definition 3.1 is a distributional generalization of both the $\nu$-strict separation and the $\alpha$-good neighborhood property from Balcan et al. [4, 5]. $\alpha, \nu$ can be viewed as noise parameters indicating the proportion of data susceptible to erroneous behavior. Next, we assume the following mild boundedness conditions on the population distribution and outcome regression function.

**Assumption A2.** $\|\mu\|_2, \|\widehat{\mu}\|_2 \leq B$ *for some finite constant $B$ a.s. $[\mathbb{P}]$.*

**Assumption A3.** $\mathbb{P}_{\alpha,\nu}$ *in Definition 3.1 has a bounded Lebesgue density.*

In the next theorem, we analyze the inductive version of the robust hierarchical clustering [5, Algorithm 2] in the causal clustering framework. We prove that when the data satisfies the good neighborhood properties, the algorithm achieves small error on the entire data set, requiring only a small random sample whose size is independent of that of the entire data set.

**Theorem 3.2.** *Suppose that* $\mathsf{U}^N$ *consists of $N$ i.i.d samples from $\mathbb{P}_{\alpha,\nu}$ that satisfies the $(\alpha,\nu)$-good neighborhood property in Definition 3.1. For $n \ll N$, consider a random subset $\mathsf{U}^n = \{\mu_{(1)}, \ldots, \mu_{(n)}\} \subset \mathsf{U}^N$ and its estimates $\widehat{\mathsf{U}}^n = \{\widehat{\mu}_{(1)}, \ldots, \widehat{\mu}_{(n)}\}$ in which clustering to be performed. Let $\gamma = \sum_{a \in \mathcal{A}} \|\widehat{\mu}_a - \mu_a\|_\infty$, and for any $\delta_N \in (0,1)$, define*

$$\alpha' = \alpha + O\left(\sqrt{\frac{1}{N} \log \frac{1}{\delta_N}}\right), \quad \nu' = \nu + O\left(\sqrt{\frac{1}{N} \log \frac{1}{\delta_N}}\right),$$

$$\varepsilon = O\left(\gamma + \frac{1}{N} \log\left(\frac{1}{\delta_N}\right) + \sqrt{\frac{\gamma}{N} \log\left(\frac{1}{\gamma}\right)}\right).$$

*Then under Assumptions A1,A2, and A3, as long as the smallest target cluster has size greater than $12(\nu' + \alpha' + \varepsilon)N$, the inductive robust hierarchical clustering [5] on $\widehat{\mathsf{U}}^n$ with $n = \Theta\left(\frac{1}{\min(\alpha'+\varepsilon,\nu')} \log \frac{1}{\delta \min(\alpha'+\varepsilon,\nu')}\right)$ produces a hierarchy with a pruning that has error at most $\nu' + \delta$ with respect to the true target clustering with probability at least $1 - \delta - 2\delta_N$.*

The main implication of Theorem 3.2 is that, in essence, the natural misclassification error $\alpha$ from the $\alpha$-good neighborhood property has increased by $O_{\mathbb{P}}(\sum_{a \in \mathcal{A}} \|\widehat{\mu}_a - \mu_a\|_\infty)$ due to the costs associated with causal clustering.

## 4 Density-based Causal Clustering

The idea of density-based clustering was initially proposed as an effective algorithm for clustering large-scale, noisy datasets [17, 24]. The density-based methods work by identifying areas of high point concentration as well as regions of relative sparsity or emptiness. It offers distinct advantages over other clustering techniques due to their adeptness in handling noise and capability to find clusters of arbitrary sizes and shapes. Further, it does not require a-priori specification of number of clusters. Here, we focus on the level-set approach [see 50, and the references therein].

With a slight abuse of notation, we let $P$ be the probability distribution of $\mu$ to distinguish it from the observational distribution $\mathbb{P}$, and $p$ be the corresponding Lebesgue density. We also let $K$ denote a valid *kernel*, i.e., a nonnegative function satisfying $\int K(u)du = 1$. We construct the oracle kernel density estimator $\widetilde{p}_h$ with the bandwidth $h > 0$ as

$$\widetilde{p}_h(\mu') = \frac{1}{n} \sum_{i=1}^{n} \frac{1}{h^q} K\left(\frac{\mu_{(i)} - \mu'}{h}\right),$$

for $\forall \mu' \in \mathbb{R}^q$. Then we define an average oracle kernel density estimator by $\mathbb{E}(\widetilde{p}_h) \equiv p_h$ and the corresponding upper level set by $L_{h,t} = \{\mu : p_h(\mu) > t\}$. Suppose that for each $t$, $L_{h,t}$ can be decomposed into finitely many disjoint sets: $L_{h,t} = C_1 \cup \cdots \cup C_{l_t}$. Then $\mathcal{C}_t = \{C_1, ..., C_{l_t}\}$ is the *level set clusters* of our interest at level $t$.

With regard to the analysis of topological properties of the distribution $P$, the upper level set of $p_h$ plays a role akin to that of the upper level set of the true density $p$, yet it presents various advantages, as indicated in previous studies [e.g., 19, 37, 50, 60]; $p_h$ is well-defined even when $p$ is not, $p_h$ provides simplified topological information, and the convergence rate of the kernel density estimator with respect to $p_h$ is faster than with $p$. For such reasons, we typically target the level set $L_{h,t}$ induced from $p_h$ in lieu of that from $p$ [see, e.g., 38, Section 2].

When each $\mu_{(i)}$ is known (or has it been observed), the level sets could be estimated by computing $\widetilde{L}_{h,t} = \{\mu : \widetilde{p}_h(\mu) > t\}$. Specifically, for each $t$ we let $\widetilde{\mathcal{W}}_t = \{\mu : \widetilde{p}_h(\mu) > t\}$, and construct a graph $G_t$ where each $\mu_{(i)} \in \widetilde{\mathcal{W}}_t$ is a vertex and there is an edge between $\mu_{(i)}$ and $\mu_{(j)}$ if and only if $\|\mu_{(i)} - \mu_{(j)}\|_2 \leq h$. Then the clusters at level $t$ are estimated by taking the connected components of the graph $G_t$, which is referred to as a *Rips graph*. *Persistent homology* measures how the topology of $R_t$ varies by the value of $t$. See, for example, Bobrowski et al. [7], Fasy et al. [19], Kent et al. [36] more information on the algorithm and its theoretical features.

However, in our causal clustering framework, the oracle kernel density estimator $\widetilde{p}_h$ is not computable since we do not observe each $\mu_{(i)}$. Thus we construct a plug-in version of the kernel density estimator:

$$\widehat{p}_h(\mu') = \frac{1}{n}\sum_{i=1}^{n}\frac{1}{h^q}K\left(\frac{\widehat{\mu}_{(i)}-\mu'}{h}\right),$$

with estimates $\{\widehat{\mu}_{(i)}\}$. Then we target the corresponding level set $\widehat{L}_{h,t} = \{\mu : \widehat{p}_h(\mu) > t\}$. To account for the added complications in estimating $\widehat{L}_{h,t}$, we introduce the following regularity conditions on the kernel $K$, along with the bounded-density condition from Assumption A3 on the distribution $P$.

**Assumption A3′.** *p is bounded a.s.* $[P]$.

**Assumption A4.** *The kernel function $K$ has a support on $\overline{\mathbb{B}(0,1)}$. Moreover, it is Lipschitz continuous with constant $M_K$, i.e., for all $x, y \in \mathbb{R}^q$, $|K(x) - K(y)| \le M_K \|x - y\|_2$.*

The Hausdorff distance is a common way of measuring difference between two sets that are embedded in the same space. In what follows, we define the Hausdorff distance for any two subsets in Euclidean space.

**Definition 4.1** (Hausdorff Distance). *Consider sets $S_1, S_2 \subset \mathbb{R}^q$. We define the Hausdorff distance $H(S_1, S_2)$ as*

$$H(S_1, S_2) = \max\left\{\sup_{x\in S_1}\inf_{y\in S_2}\|x-y\|_2, \sup_{y\in S_2}\inf_{x\in S_1}\|x-y\|_2\right\}.$$

Note that the Hausdorff distance can be equivalently defined as

$$H(S_1, S_2) = \inf\left\{\epsilon \ge 0 : S_1 \subset S_{2,\epsilon} \text{ and } S_2 \subset S_{1,\epsilon}\right\},$$

where for $i = 1, 2$, $S_{i,\epsilon} := \{y \in \mathbb{R}^q : \text{there exists } x \in S_i \text{ with } \|x - y\|_2 \le \epsilon\}$.

To estimate the target level set $L_{t,h} = \{p_h > t\}$ using the estimator $\widehat{L}_{t,h} = \{\widehat{p}_h > t\}$, we normally assume that the function difference $\|\widehat{p}_h - p_h\|_\infty$ is small. To apply this condition to the set difference $H(L_{t,h}, \widehat{L}_{t,h})$, we have to ensure that the target level set $L_{t,h}$ does not change drastically when the level $t$ perturbs. We formalize this notion as follows.

**Definition 4.2** (Level Set Stability). *We say that the level set $L_{t,h} = \{w \in \mathbb{R}^q : p_h(w) > t\}$ is stable if there exists $a > 0$ and $C > 0$ such that, for all $\delta < a$,*

$$H(L_{t-\delta,h}, L_{t+\delta,h}) \le C\delta.$$

The next theorem shows provided that the target level set $L_{h,t}$ is stable in the sense of Definition 4.2, our level set estimator $\widehat{L}_{h,t}$ is close to the target level set $L_{h,t}$ in the Hausdorff distance.

**Theorem 4.1.** *Suppose that $L_{h,t}$ is stable and let $H(\cdot, \cdot)$ be the Hausdorff distance between two sets. Let the bandwidth $h$ vary with $n$ such that $\{h_n\}_{n\in\mathbb{N}} \subset (0, h_0)$ and*

$$\limsup_n \frac{(\log(1/h_n))_+}{nh_n^q} < \infty.$$

*Then, under Assumptions A1, A2, A3′, and A4, we have that with probability at least $1 - \delta$,*

$$H(\widehat{L}_{h,t}, L_{h,t}) \lesssim \sqrt{\frac{(\log(1/h_n))_+ + \log(2/\delta)}{nh_n^q}} + \frac{1}{h_n^{q+1}}\min\left\{\sum_a\|\widehat{\mu}_a - \mu_a\|_1 + \sqrt{\frac{\log(2/\delta)}{n}}, h_n\right\}.$$

The above theorem ensures that the estimated level sets in the causal clustering framework do not significantly deviate from $L_{h,t}$, provided that the error of $\widehat{\mu}_a$ remains small. As a consequence, we show that causal clustering may also be accomplished via level-set density-based clustering, albeit at the expense of estimating the nuisance regression functions for the outcome process. The bandwidth $h$ may be selected either by minimizing the error bounds derived from Theorem 4.1 or by employing data-driven methodologies [e.g., 38, Remark 5.1].

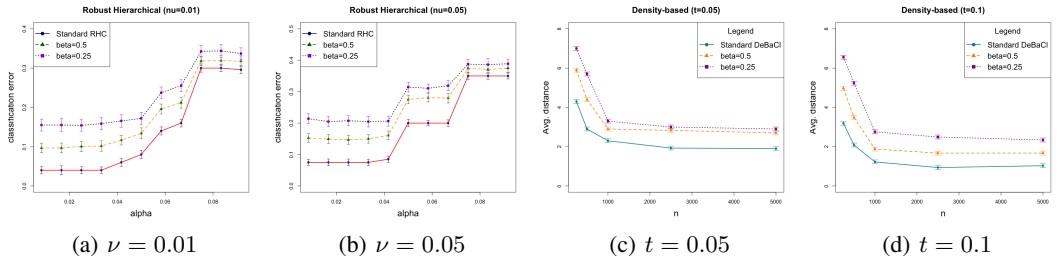

(a) $\nu = 0.01$    (b) $\nu = 0.05$    (c) $t = 0.05$    (d) $t = 0.1$

Figure 2: (a), (b): The y-axis represents classification error from hierarchical (causal) clustering, where we fix $\nu = 0.01, 0.1$ and vary $\alpha$. (c), (d): The y-axis represents the average of $H(\widehat{L}_{h,t}, L_{h,t})$ from density-based (causal) clustering, where we fix $t = 0.05, 0.1$ and vary $n$.

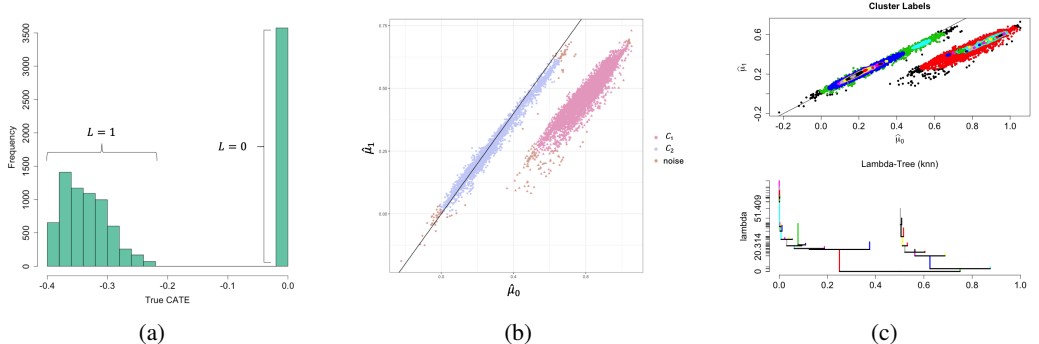

(a)       (b)       (c)

Figure 3: (a) Histogram of the true CATE in the test set. In the original study [46], individuals with zero treatment effects are assigned to the label $L = 0$. (b) The result of density-based causal clustering. Units in Cluster C1 appear to have higher baseline risk ($\mu_0$). (c) We observe that points in Clusters C1 and C2 are more concentrated around the right upper area (larger $\mu_0, \mu_1$) and the lower left area (smaller $\mu_0, \mu_1$), respectively.

## 5 Empirical Analyses

### 5.1 Simulation Study

Here, we explore finite-sample properties of our proposed plug-in procedures via simulation. In particular, we investigate the effect of nuisance estimation on the performance of causal clustering to empirically validate our theoretical findings in Sections 3 and 4.

For hierarchical causal clustering, we use the simulation setup akin to that of Kim et al. [39]. Letting $n = 2500$, we randomly pick 10 points in a bounded hypercube $[0,1]^3$: $\{c_1^*, ..., c_{10}^*\}$, and assign roughly $n/10$ points following truncated normal distribution to each Voronoi cell associated with $c_j^*$; these are our $\{\mu_{(i)}\}$. Next, we let $\widehat{\mu}_a = \mu_a + \xi$ with $\xi \sim N(0, n^{-\beta})$, which ensures that $\|\widehat{\mu}_a - \mu_a\| = O_{\mathbb{P}}(n^{-\beta})$. Following Balcan et al. [5], by repeating simulations 100 times, we compute classification error as a proxy of the clustering performance using different values of parameter $\alpha$ fixing the value of $\nu$ and $\beta$. The results are presented in Figure 2 (a) & (b) with standard deviation error bars. The simulation result supports our finding in Theorem 3.2, indicating that the price we pay for the proposed hierarchical causal clustering is inflated $\alpha$ due to the nuisance estimation error.

For density-based causal clustering, we utilize the toy example from Fasy et al. [18], originally used to illustrate the cluster tree. We consider a mixture of three Gaussians in $\mathbb{R}^2$. Then, roughly $n/3$ points for each of the three clusters are generated, which are our $\{\mu_{(i)}\}$. Similarly as before, we let $\widehat{\mu}_a = \mu_a + \xi$ with $\xi \sim N(0, n^{-\beta})$. Next, letting $h = 0.01$, we compute $\widetilde{p}_h$ and $\widehat{p}_h$, and the corresponding level sets $L_{h,t}$ and $\widehat{L}_{h,t}$ for different values of $t$. For each $n$, we calculate the mean Hausdorff distance between $\widehat{L}_{h,t}$ and $L_{h,t}$ through 100 repeated simulations, and present the results

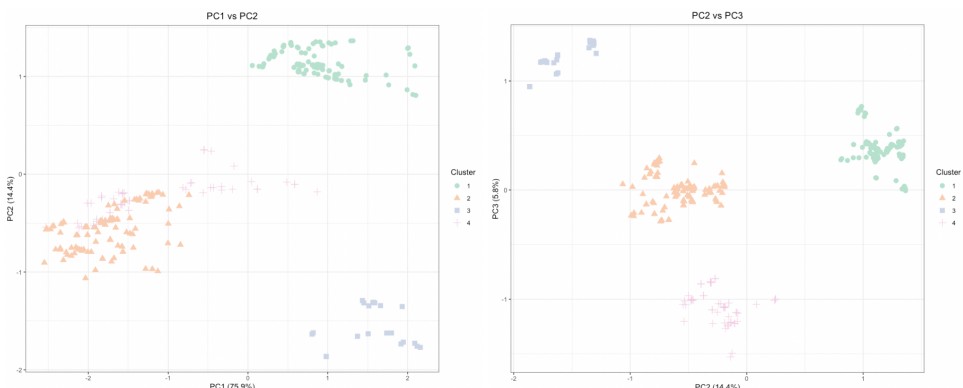

Figure 4: The estimated causal clusters on two principal-component hyperplanes with axes representing the first and second, second and third principal components in the conditional counterfactual mean vector space, respectively.

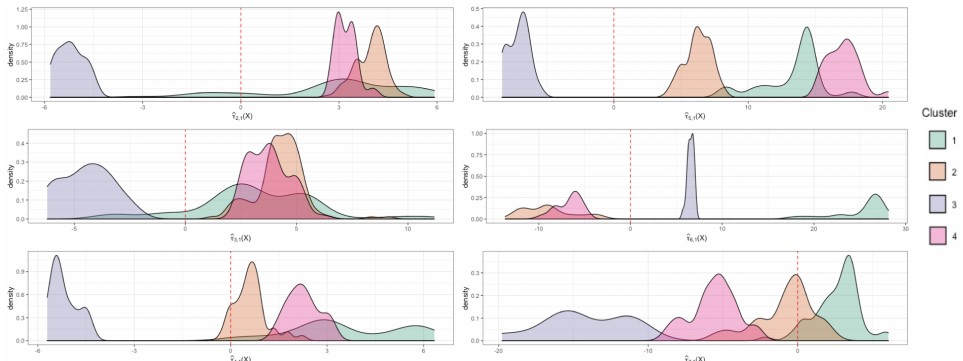

Figure 5: The density plots of the pariwise CATE of six other education levels relative to the doctoral degree across clusters. We observe a substantial degree of effect heterogeneity. The red dashed vertical lines denote the zero CATE.

in Figure 2 (c) & (d) with error bars. Again, the results corroborate the conclusion from Theorem 4.1 that the cost of causal clustering is associated with the nuisance estimation error.

## 5.2 Case Study

In this section, we illustrate our method through two case studies. We use semi-synthetic data on the voting study and real-world data on employment projections.

**Voting study**. Nie and Wager [46] considered a dataset on the voting study originally used by Arceneaux et al. [2]. They generated synthetic treatment effects to render discovery of heterogeneous treatment effects more challenging. We use the same setup as Nie and Wager [46, Chapter 2], where we have binary treatments, binary outcomes, and 11 pretreatment covariates. While Nie and Wager [46] specifically focused on accurate estimation of the CATE, here we aim to illustrate how the proposed method can be used to uncover an intriguing subgroup structure. We randomly chose a training set of size $13000$ and a test set of size $10000$ from the entire sample. Then we estimate $\{\widehat{\mu}_{(i)}\}$ using the cross-validation-based Super Learner ensemble [54] to combine regression splines, support vector machine regression, and random forests on the training sample, and perform the density-based causal clustering on the test sample using `DeBaCl` function in `TDA` R package [18].

In Figure 3-(b), we see two clusters in our conditional counterfactual mean vector space that are clearly separable from each other, one with nearly zero subgroup effect (Cluster C2) and the other with negative effect (Cluster C1). They correspond to the two largest branches at the bottom of the tree (Figure 3-(c)). Roughly $4\%$ of the points are classified as noise. Interestingly, units in Cluster C1 appear to have higher baseline risk $\mu_0$ than Cluster C2. This is indeed more clearly noticeable in

Figure 3-(c); Clusters C1 and C2 have a higher concentration of units in the right upper area (larger $\mu_0, \mu_1$) and the lower left area (smaller $\mu_0, \mu_1$).

**Employment projection data**.

The dataset, obtained from the US Bureau of Labor Statistics (BLS), provides projected employment by occupation. Specifically, the dataset consists of projected 10-year employment changes (2018-2028) computed from the BLS macroeconomic model across various occupations. We have eight education levels (No formal education, High school, Bachelor's degree, etc.). Here, we aim to uncover subgroup structure in the effects of entry-level education on projected employment. Our data also include four covariates: baseline employment in 2018, median annual wage in 2019, work experience, and on-the-job training.

Again we randomly split the data into two independent sets and use the super learner ensemble to estimate the nuisance regression functions. We then find clusters using robust hierarchical causal clustering described in Section 3. Since we have multi-level treatments this time ($q = 8$), for ease of visualization, in Figure 4 we present the resulting clusters in two-dimensional hyperplanes with axes representing the first and second, second and third principal components, respectively. We also present the density plots for some of the pairwise CATEs across clusters in Figure 5.

In Figure 4, we observe four distinct clusters which are quite clearly separable from each other on the principal component hyperplanes. It appears that some clusters show considerably different effects from the others (e.g., Cluster 3), as shown in Figure 5. Our findings indicate a substantial heterogeneity in the effects of entry-level education on projected employment.

## 6 Discussion

Causal clustering is a new approach for studying treatment effect heterogeneity that draws on cluster analysis tools. In this work, we expanded upon this framework by integrating widely-used hierarchical and density-based clustering algorithms, where we presented and analyzed the simple and readily implementable plug-in estimators. Importantly, as we do not impose any restrictions on the outcome process, the proposed methods offer novel opportunities for clustering with generic unknown pseudo outcomes.

There are some caveats and limitations which should be addressed. First, causal clustering plays a more descriptive and discovery-based than prescriptive role compared to other approaches. It enables efficient discovery of subgroup structures and intriguing subgroup features as illustrated in our case studies, yet will likely be less useful for informing specific treatment decisions. Understanding this trade-off is thus important, and we recommend using our methods in conjunction with other approaches. Nonetheless, the clustering outputs could be potentially utilized as an useful input for subsequent learning tasks, such as precision medicine or optimal policy. Next, our theoretical findings show that when the nuisance regression functions $\{\mu_a\}$ are modeled nonparametrically, the clustering performance essentially inherits from that of $\{\widehat{\mu}_a\}$. The convergence rate of $\widehat{\mu}_a$ can be arbitrarily slow as the dimension of the covariate space increases. Kim et al. [39] addressed this issue by developing an efficient semiparametric estimator that achieves the second-order bias, and so can attain fast rates even in high-dimensional covariate settings [34]. In future work, we aim to develop more efficient semiparametric estimators for hierarchical and density-based causal clustering. Extension to other robust clustering methods, such as hierarchical density-based clustering [10], would be a promising direction for future research as well. Lastly, our proposed methods are currently limited to the standard identification strategy under the no-unmeasured-confounding assumption, which is typically vulnerable to criticism [e.g., 28, Chapter 12]. To widen the breadth of the causal clustering framework, we will also be exploring extensions to other identification strategies, such as instrumental variable, mediation, and proximal causal learning.

## 7 Broader Impact

The proposed method provides a general framework for causal clustering that is not specifically tailored to any particular application, thereby reducing the potential for unintended societal or ethical impacts. Nonetheless, it is important to note that the identified subgroup structure was constructed entirely based on treatment effect similarity, without accounting for fairness or bias.

## 8    Acknowledgements

This work was partially supported by a Korea University Grant (K2407471) and the National Research Foundation of Korea (NRF) grant funded by the Korea governement (MSIT)(No. NRF-2022M3J6A1063595). This work was also partially supported by Institute of Information & communications Technology Planning & Evaluation (IITP) grant funded by the Korea government(MSIT) [NO.RS-2021-II211343, Artificial Intelligence Graduate School Program (Seoul National University)] and the New Faculty Startup Fund from Seoul National University. Part of this work was completed while Kwangho Kim was a Ph.D. student at Carnegie Mellon University.

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

# Appendix

## A  Robust Hierarchical Clustering

This section summarizes definitions and theoretical results in Balcan et al. [5].

We first define a clustering problem $(\mathsf{U}, l)$ as follows. Assume we have a data set $\mathsf{U}$ of $N$ objects. Each point $\mu' \in \mathsf{U}$ has a true cluster label $l(\mu')$ in $\{C_1, \ldots, C_k\}$. Further we let $C(\mu')$ denote a cluster corresponding to the label $l(\mu')$, and $n_{C(\mu')}$ denote the size of the cluster $C(\mu')$.

The good-neighborhood property in Definition A.1 is a generalization of both the $\nu$-strict separation and the $\alpha$-good neighborhood property in Balcan et al. [5]. It roughly means that after a portion of points are removed, each point might allow some bad immediate neighbors but most of the immediate neighbors are good. $\alpha$, $\nu$ can be viewed as noise parameters indicating the proportion of data susceptible to erroneous behavior.

**Definition A.1** (Property 3 in [5]). *Suppose a clustering problem $(\mathsf{U}, l)$ with $|\mathsf{U}| = N$, and a similarity function $d : \mathsf{U} \times \mathsf{U} \to \mathbb{R}$. We say the similarity function $d$ satisfies $(\alpha, \nu)$-good neighborhood property for the clustering problem $(\mathsf{U}, l)$, if there exists $\mathsf{U}' \subset \mathsf{U}$ of size $(1 - \nu)N$ so that for all points $\mu' \in \mathsf{U}'$ we have that all but $\alpha N$ out of their $n_{C(\mu') \cap \mathsf{U}'}$ nearest neighbors in $S'$ belong to the cluster $C(\mu')$.*

In the inductive setting, Algorithm 2 in Balcan et al. [5] uses a random sample over the data set and generates a hierarchy over this sample, and also implicitly represents a hierarchy over the entire data set. When the data satisfies the good neighborhood properties, Algorithm 2 in [5] achieves small error on the entire data set, requiring only a small sample size independent of that of the entire data set, as in Theorem A.1.

**Theorem A.1** (Theorem 11 in [5]). *Let $d$ be a symmetric similarity function satisfying the $(\alpha, \nu)$-good neighborhood for the clustering problem $(\mathsf{U}, l)$. As long as the smallest target cluster has size greater than $12(\nu + \alpha)N$, then Algorithm 2 in [5] with parameters $n = \Theta\left(\frac{1}{\min(\alpha, \nu)} \log \frac{1}{\delta \min(\alpha, \nu)}\right)$ produces a hierarchy with a pruning that has error at most $\nu + \delta$ with respect to the true target clustering with probability at least $1 - \delta$.*

## B  Proofs

Throughout the development, we let $\mathbb{P}$ denote the conditional expectation given the sample operator $\hat{f}$, as in $\mathbb{P}(\hat{f}) = \int \hat{f}(z) d\mathbb{P}(z)$. Notice that $\mathbb{P}(\hat{f})$ is random only if $\hat{f}$ depends on samples, in which case $\mathbb{P}(\hat{f}) \neq \mathbb{E}(\hat{f})$. Otherwise $\mathbb{P}$ and $\mathbb{E}$ can be used exchangeably. For example, if $\hat{f}$ is constructed on a separate (training) sample $\mathsf{D}^n = (Z_1, ..., Z_n)$, then $\mathbb{P}\left\{\hat{f}(Z)\right\} = \mathbb{E}\left\{\hat{f}(Z) \mid \mathsf{D}^n\right\}$ for a new observation $Z \sim \mathbb{P}$. Lastly, we let $d_2 : \mathbb{R}^q \times \mathbb{R}^q \to \mathbb{R}$ be the Euclidean distance on $\mathbb{R}^q$, i.e.,

$$d_2(x, y) := \|x - y\|_2.$$

We present the following basic lemma which we will use throughout the proofs.

**Lemma B.1.** *Under Assumption A1,*

*(a) The conditional expectation of $\|\widehat{\mu} - \mu\|_2$ is bounded by the estimation error of $\widehat{\mu}_a$: i.e.,*

$$\mathbb{P}\left(\|\widehat{\mu} - \mu\|_2\right) \leq \sum_a \|\widehat{\mu}_a - \mu_a\|_{\mathbb{P},1}. \tag{2}$$

*(b) Under Assumption A2, for $\delta \in (0, 1)$, $\frac{1}{n} \sum_{i=1}^{n} \left\|\widehat{\mu}_{(i)} - \mu_{(i)}\right\|_2$ can be bounded with probability at least $1 - \delta$ as*

$$\frac{1}{n} \sum_{i=1}^{n} \left\|\widehat{\mu}_{(i)} - \mu_{(i)}\right\|_2 \leq \sum_a \|\widehat{\mu}_a - \mu_a\|_1 + B\sqrt{\frac{\log(1/\delta)}{n}}. \tag{3}$$

*Proof of Lemma B.1.* (a) It is immediate to see that

$$\mathbb{P}\left[\left\|\widehat{\mu}_{(i)} - \mu_{(i)}\right\|_2\right] = \mathbb{P}\left[\sqrt{\sum_a (\widehat{\mu}_a(X) - \mu_a(X))^2}\right]$$

$$\leq \sum_a \mathbb{P}\left[|\widehat{\mu}_a(X) - \mu_a(X)|\right]$$

$$= \sum_a \|\widehat{\mu}_a - \mu_a\|_{\mathbb{P},1}.$$

(b) Noting that Assumption A2 implies $0 \leq \left\|\widehat{\mu}_{(i)} - \mu_{(i)}\right\|_2 \leq \sqrt{2}B$ a.s., by Hoeffding's inequality we get

$$\mathbb{P}\left(\frac{1}{n}\sum_{i=1}^n \left\|\widehat{\mu}_{(i)} - \mu_{(i)}\right\|_2 - \mathbb{P}\left[\left\|\widehat{\mu}_{(i)} - \mu_{(i)}\right\|_2\right] > t\right) \leq \exp\left(-\frac{nt^2}{B^2}\right).$$

Hence for any $\delta > 0$, applying $t = B\sqrt{\frac{\log(1/\delta)}{n}}$ gives

$$\mathbb{P}\left(\frac{1}{n}\sum_{i=1}^n \left\|\widehat{\mu}_{(i)} - \mu_{(i)}\right\|_2 \leq \mathbb{P}\left[\left\|\widehat{\mu}_{(i)} - \mu_{(i)}\right\|_2\right] + B\sqrt{\frac{\log(1/\delta)}{n}}\right) \geq 1 - \delta.$$

Then applying (2) gives

$$\mathbb{P}\left(\frac{1}{n}\sum_{i=1}^n \left\|\widehat{\mu}_{(i)} - \mu_{(i)}\right\|_2 \leq \sum_a \|\widehat{\mu}_a - \mu_a\|_1 + B\sqrt{\frac{\log(1/\delta)}{n}}\right) \geq 1 - \delta.$$

$\square$

## Proof of Proposition 3.1

We rewrite Proposition 3.1 with detailed constants relation.

**Proposition B.2.** *Let $D$ denote the single, average, or complete linkage between sets of points, induced by the distance function such that $d(x, y) \leq \mathsf{C}\|x - y\|_1$ for some constant $\mathsf{C} > 0$. Then under Assumption A1, for any two sets $S_1, S_2$ in $\{\mu_{(i)}\}$ and their estimates $\widehat{S}_1, \widehat{S}_2$ with $\{\widehat{\mu}_{(i)}\}$,*

$$\left|D(S_1, S_2) - D(\widehat{S}_1, \widehat{S}_2)\right| \leq 2\mathsf{C}\sum_{a \in \mathcal{A}} \|\widehat{\mu}_a - \mu_a\|_\infty.$$

*Proof of Proposition B.2.* Recall that we are given a pair of points $s_1 = (\mu_1(X_1), ..., \mu_q(X_1))$, $s_2 = (\mu_1(X_2), ..., \mu_q(X_2))$, and their estimates $\widehat{s}_1 = (\widehat{\mu}_1(X_1), ..., \widehat{\mu}_q(X_1))$, $\widehat{s}_2 = (\widehat{\mu}_1(X_2), ..., \widehat{\mu}_q(X_2))$ for $\forall X_1, X_2 \in \mathcal{X}$. To prove the theorem, first we upper bound the maximum discrepancy between $d(s_1, s_2)$ and $d(\widehat{s}_1, \widehat{s}_2)$. Since our distance function satisfies

$$d(x, y) \leq \mathsf{C}\|x - y\|_1$$

for any $x, y \in \mathbb{R}^p$, we may get

$$d(s_1, s_2) - d(\widehat{s}_1, \widehat{s}_2)$$
$$\leq \mathsf{C}\|\mu(X_1) - \mu(X_2) - \{\widehat{\mu}(X_1) - \widehat{\mu}(X_2)\}\|_1$$
$$\leq \mathsf{C}\sum_{j=1}^2 \sum_{a \in \mathcal{A}} |\widehat{\mu}_a(X_j) - \mu_a(X_j)|$$
$$\leq 2\mathsf{C}\sum_{a \in \mathcal{A}} \|\widehat{\mu}_a - \mu_a\|_\infty.$$

where the first inequality follows by $\|x\|_1 - \|y\|_1 \leq \|x - y\|_1$.

Let $(s_1{}^*, s_2{}^*) = \underset{s_1 \in S_1, s_2 \in S_2}{\arg\min}\, d(s_1, s_2)$ and $\widehat{s}_1^*, \widehat{s}_2^*$ denote their estimates. Then by the definition of single linkage we have

$$
\begin{aligned}
\left| D(S_1, S_2) - D(\widehat{S}_1, \widehat{S}_2) \right| &= \left| \min_{\widehat{a} \in \widehat{A}, \widehat{b} \in \widehat{B}} d(\widehat{s}_1^*, \widehat{s}_2^*) - d(s_1^*, s_2^*) \right| \\
&\leq | d(\widehat{s}_1^*, \widehat{s}_2^*) - d(s_1^*, s_2^*) | \\
&\leq 2\mathsf{C} \sum_{a \in \mathcal{A}} \| \widehat{\mu}_a - \mu_a \|_\infty.
\end{aligned}
$$

The same logic applies to the complete and average linkages.

$\square$

**Proof of Theorem 3.2**

Recall that we let $\mu$ denote a point in the conditional counterfactual mean vector space given $\mathcal{X}, \mathcal{A}$. We also use the notation $\mathsf{U}^N := \{\mu_{(1)}, \ldots, \mu_{(N)}\}$, where $\mu_{(i)}$'s are i.i.d. samples from $\mathbb{P}$. Further by Assumption A3, we assume that every distribution satisfying the good neighborhood property in Definition 3.1 has a density bounded by $p_\mu < \infty$. We begin with introducing some useful lemmas before proving our main theorem.

**Lemma B.3.** *Under Assumptions A2, A3,*

$$
\sup_{w \in \mathbb{R}^q, r > 0} \mathbb{P}\left( \mu \in \mathbb{B}(w, r+s) \backslash \mathbb{B}(w, r) \right) \leq \mathsf{C}_1 s,
$$

*where $\mathsf{C}_1$ is some constant that depends on $p_\mu$, $B$, and $q$.*

*Proof.* Let $\lambda_q$ be the $q$-dimensional Lebesgue measure. By Assumption (A2), $\mathrm{supp}(p_\mu) \subset [-2B, 2B]^q$, and hence for any $w \in \mathbb{R}^q$ and $r, s > 0$,

$$
\lambda_q \left( (\mathbb{B}(w, r+s) \backslash \mathbb{B}(w, r)) \cap \mathrm{supp}(p_\mu) \right) \leq \lambda_q \left( (\mathbb{B}(w, r+s) \backslash \mathbb{B}(w, r)) \cap [-2B, 2B]^q \right).
$$

Now, we bound $\lambda_{q-1}(\partial\mathbb{B}(w, t) \cap [-2B, 2B]^q)$ for any $t \in \mathbb{R}$. First, note that for any $u \geq 0$, by considering that the map $\varphi : \partial\mathbb{B}(w, t) \cap [-2B, 2B]^q \to \partial\mathbb{B}(w, t+u) \cap [-2B-u, 2B+u]^q$ by $\varphi(w + tv) = w + (t+u)v$ for unit vector $v$ satisfies $\|\varphi(x) - \varphi(y)\| \geq \|x - y\|$, we have

$$
\lambda_{q-1}(\partial\mathbb{B}(w, t) \cap [-2B, 2B]^q) \leq \lambda_{q-1}(\partial\mathbb{B}(w, t+u) \cap [-2B-u, 2B+u]^q).
$$

And hence

$$
\begin{aligned}
\frac{2B}{q} &\lambda_{q-1}(\partial\mathbb{B}(w, t) \cap [-2B, 2B]^q) \\
&= \int_0^{\frac{2B}{q}} \lambda_{q-1}(\partial\mathbb{B}(w, t) \cap [-2B, 2B]^q) du \\
&\leq \int_0^{\frac{2B}{q}} \lambda_{q-1}\left( \partial\mathbb{B}(w, t+u) \cap [-2B-u, 2B+u]^q \right) du \\
&\leq \int_0^{\frac{2B}{q}} \lambda_{q-1}\left( \partial\mathbb{B}(w, t+u) \cap \left[ -2(1 + \frac{1}{q})B, 2(1 + \frac{1}{q})B \right]^q \right) du \\
&= \lambda_q \left( \mathbb{B}(w, t+B) \backslash \mathbb{B}(w, t)) \cap \left[ -2(1 + \frac{1}{q})B, 2(1 + \frac{1}{q})B \right]^q \right) \\
&\leq \lambda_q \left( \left[ -2(1 + \frac{1}{q})B, 2(1 + \frac{1}{q})B \right]^q \right) \leq e4^q B^q,
\end{aligned}
$$

and hence

$$
\lambda_{q-1}(\partial\mathbb{B}(w, t) \cap [-2B, 2B]^q) \leq e2^{2q-1} B^{q-1} q.
$$

Then $\lambda_q\left((\mathbb{B}(w, r + s)\backslash\mathbb{B}(w, r)) \cap [-2B, 2B]^q\right)$ is bounded as

$$\lambda_q\left((\mathbb{B}(w, r + s)\backslash\mathbb{B}(w, r)) \cap [-2B, 2B]^q\right) = \int_0^s \lambda_{q-1}(\partial\mathbb{B}(w, r + t) \cap [-2B, 2B]^q)dt$$
$$\leq \int_0^s e2^{2q-1}B^{q-1}pdt = e2^{2q-1}B^{q-1}qs.$$

And hence for all $w \in \mathbb{R}^q$ and $r > 0$, Under Assumption A3,

$$\mathbb{P}\left(\mu \in \mathbb{B}(w, r + s)\backslash\mathbb{B}(w, r)\right) \leq p_\mu \int_{(\mathbb{B}(w,r+s)\backslash\mathbb{B}(w,r))\cap\mathrm{supp}(p_\mu)} \lambda_q\,(dw)$$
$$\leq ep_\mu 2^{2q-1}B^{q-1}qs.$$

$\square$

**Lemma B.4.** *Suppose* $\mathsf{U}^N := \{\mu_{(1)}, \ldots, \mu_{(N)}\}$ *are i.i.d. samples from* $\mathbb{P}$. *With probability* $1 - \delta_N$,

$$\sup_{w\in\mathbb{R}^q,r>0}\left|\frac{\left|\mathsf{U}^N \cap (\mathbb{B}(w, r + s)\backslash\mathbb{B}(w, r))\right|}{N} - \mathbb{P}\left(\mu \in \mathbb{B}(w, r + s)\backslash\mathbb{B}(w, r)\right)\right|$$
$$\leq \mathsf{C}_2\left(\frac{1}{N}\log\left(\frac{1}{\delta_N}\right) + \sqrt{\frac{s}{N}\log\left(\frac{1}{s}\right)} + \sqrt{\frac{s}{N}\log\left(\frac{1}{\delta_N}\right)}\right),$$

*where* $\mathsf{C}_2$ *is a constant depending only on* $q$, $B$, $p_\mu$.

*Proof.* For $w \in \mathbb{R}^q$ and $r, s > 0$, let $B_{w,r,s} := \mathbb{B}(w, r + s)\backslash\mathbb{B}(w, r)$, and let $\mathcal{F}_s := \left\{\mathbb{1}_{B_{w,r,s}} : w \in \mathbb{R}^q, r > 0\right\}$. Then

$$\sup_{w\in\mathbb{R}^q,r>0}\left|\frac{\left|\mathsf{U}^N \cap (\mathbb{B}(w, r + s)\backslash\mathbb{B}(w, r))\right|}{N} - \mathbb{P}\left(\mu \in \mathbb{B}(w, r + s)\backslash\mathbb{B}(w, r)\right)\right|$$
$$= \sup_{f\in\mathcal{F}_s}\left|\frac{1}{N}\sum_{i=1}^N f(\mu_{(i)}) - \mathbb{E}\left[f(\mu_{(i)})\right]\right|.$$

Now, for $w \in \mathbb{R}^q$ and $r > 0$, let $B_{w,r} := \mathbb{B}(w, r)$ and $\tilde{B}_{w,r} := \mathbb{R}^q\backslash\mathbb{B}(w, r)$, and let $\mathcal{H} := \{B_{w,r} : w \in \mathbb{R}^q, r > 0\}$ and $\tilde{\mathcal{H}} := \left\{\tilde{B}_{w,r} : w \in \mathbb{R}^q, r > 0\right\}$. Then the VC dimension of $\mathcal{H}$ or $\tilde{\mathcal{H}}$ is no greater than $q + 2$. Therefore, let $s(\mathcal{H}, N)$ and $s(\tilde{\mathcal{H}}, N)$ be shattering number of $\mathcal{H}$ and $\tilde{\mathcal{H}}$, respectively, then by Sauer's Lemma for $N \geq q + 2$,

$$s(\mathcal{H}, N) \leq \left(\frac{eN}{q+2}\right)^{q+2} \quad \text{and} \quad s(\tilde{\mathcal{H}}, N) \leq \left(\frac{eN}{q+2}\right)^{q+2}.$$

Now, let $\mathcal{G}_s := \{B_{w,r,s} : w \in \mathbb{R}^q, r > 0\}$, then $\mathcal{G}_s \subset \left\{A \cap B : A \in \mathcal{H}, B \in \tilde{\mathcal{H}}\right\}$, and hence for $N \geq q + 2$,

$$s(\mathcal{G}_s, N) \leq s(\mathcal{H}, N)s(\tilde{\mathcal{H}}, N) \leq \left(\frac{eN}{q+2}\right)^{2q+4}.$$

Then, for $N = (2q + 4)^2$,

$$s(\mathcal{G}_s, (2q + 4)^2) \leq (2e(2q + 4))^{2q+4}$$
$$< (2^{2q+4})^{2q+4} = 2^{(2q+4)^2},$$

so VC dimension of $\mathcal{G}_s$ is bounded by $(2q+4)^2$. Then from Theorem 2.6.4 in Van Der Vaart and Wellner [55],

$$\mathcal{N}(\mathcal{F}_s, \|\cdot\|, \epsilon) \leq K(2q+4)^2 (4e)^{(2q+4)^2} \left(\frac{1}{\epsilon}\right)^{2((2q+4)^2-1)}$$

$$\leq \left(\frac{8K(q+2)e}{\epsilon}\right)^{2((2q+4)^2-1)},$$

for some universal constant $K$. Now, for all $f \in \mathcal{F}_s$, we have $\mathbb{E}f^2 \leq C_{B,p_\mu,q}s$ from Lemma B.3. Hence, by Theorem 30 in Kim et al. [37], with probability $1 - \delta_N$,

$$\sup_{f \in \mathcal{F}_s} \left| \frac{1}{N} \sum_{i=1}^{N} f(\mu_{(i)}) - \mathbb{E}\left[f(\mu_{(i)})\right] \right|$$

$$\leq C \left( \frac{\nu_q}{N} \log(2\Lambda_q) + \sqrt{\frac{\nu_q \mathsf{C}_3 s}{N} \log\left(\frac{2\Lambda_q}{\mathsf{C}_3 s}\right)} + \sqrt{\frac{\mathsf{C}_3 s \log(\frac{1}{\delta_N})}{N}} + \frac{\log(\frac{1}{\delta_N})}{N} \right),$$

where $\nu_q = 2((2q+4)^2 - 1)$ and $\Lambda_q = 8K(q+2)e$. Hence, it can be simplified as

$$\sup_{f \in \mathcal{F}_s} \left| \frac{1}{N} \sum_{i=1}^{N} f(\mu_{(i)}) - \mathbb{E}\left[f(\mu_{(i)})\right] \right| \leq \mathsf{C}_2 \left( \frac{1}{N} \log\left(\frac{1}{\delta_N}\right) + \sqrt{\frac{s}{N} \log\left(\frac{1}{s}\right)} + \sqrt{\frac{s}{N} \log\left(\frac{1}{\delta_N}\right)} \right),$$

where $\mathsf{C}_2$ is a constant depending only on $q$, $B$, $p_\mu$.

$\square$

**Corollary B.5.** *Suppose* $\mathsf{U}^N := \{\mu_{(1)}, \ldots, \mu_{(N)}\}$ *are i.i.d. samples from* $\mathbb{P}$. *Under Assumption A3, with probability* $1 - \delta_N$, *we have*

$$\sup_{w \in \mathbb{R}^q, r > 0} \frac{\left| \mathsf{U}^N \cap (\mathbb{B}(w, r+s) \backslash \mathbb{B}(w, r)) \right|}{N} \leq \mathsf{C}_3 \left( s + \frac{1}{N} \log\left(\frac{1}{\delta_N}\right) + \sqrt{\frac{s}{N} \log\left(\frac{1}{s}\right)} \right),$$

*where* $\mathsf{C}_3$ *is a constant depending only on* $q$, $B$, $p_\mu$.

*Proof.*

$$\sup_{w \in \mathbb{R}^q, r > 0} \left| \mathsf{U}^N \cap (\mathbb{B}(w, r+s) \backslash \mathbb{B}(w, r)) \right|$$

$$\leq \sup_{w \in \mathbb{R}^q, r > 0} \mathbb{P}\left( \mu \in \mathbb{B}(w, r+s) \backslash \mathbb{B}(w, r) \right)$$

$$+ \sup_{w \in \mathbb{R}^q, r > 0} \left| \frac{\left| \mathsf{U}^N \cap (\mathbb{B}(w, r+s) \backslash \mathbb{B}(w, r)) \right|}{N} - \mathbb{P}\left( \mu \in \mathbb{B}(w, r+s) \backslash \mathbb{B}(w, r) \right) \right|.$$

Then from Lemma B.3 and B.4, with probability $1 - \delta_N$,

$$\sup_{w \in \mathbb{R}^q, r > 0} \left| \mathsf{U}^N \cap (\mathbb{B}(w, r+s) \backslash \mathbb{B}(w, r)) \right|$$

$$\leq \mathsf{C}_1' s + \mathsf{C}_2 \left( \frac{1}{N} \log\left(\frac{1}{\delta_N}\right) + \sqrt{\frac{s}{N} \log\left(\frac{1}{s}\right)} + \sqrt{\frac{s}{N} \log\left(\frac{1}{\delta_N}\right)} \right)$$

$$\leq \mathsf{C}_1' s + \mathsf{C}_2 \left( \frac{1}{N} \log\left(\frac{1}{\delta_N}\right) + \sqrt{\frac{s}{N} \log\left(\frac{1}{s}\right)} + \frac{1}{2}\left( s + \frac{1}{N} \log\left(\frac{1}{\delta_N}\right) \right) \right)$$

$$\leq \mathsf{C}_3 \left( s + \frac{1}{N} \log\left(\frac{1}{\delta_N}\right) + \sqrt{\frac{s}{N} \log\left(\frac{1}{s}\right)} \right),$$

where $\mathsf{C}_3 = \max\left\{ \mathsf{C}_1' + \frac{1}{2}\mathsf{C}_2, \frac{3}{2}\mathsf{C}_2 \right\}$.

$\square$

**Lemma B.6.** *Suppose* $\mathsf{U}^N := \{\mu_{(1)}, ..., \mu_{(N)}\}$ *are i.i.d samples from the mixture distribution* $\mathbb{P}_{\alpha,\nu}$ *defined in Definition 3.1. Then with probability* $1 - \delta_N$, *the distance* $d_2$ *satisfies* $(\alpha', \nu')$-*good neighborhood property for the clustering problem* $(\mathsf{U}^N, l)$, *where*

$$\alpha' = \alpha + O\left(\sqrt{\frac{1}{N}\log\frac{1}{\delta_N}}\right) \quad and \quad \nu' = \nu + O\left(\sqrt{\frac{1}{N}\log\frac{1}{\delta_N}}\right).$$

*Proof.* For any $\delta_N \in (0, 1)$, by Hoeffding's inequality we have

$$\frac{1}{N}\sum_{i=1}^{N}\mathbb{1}\left\{\mu_{(i)} \sim \mathbb{P}_{\text{noise}}\right\} \geq \nu + \sqrt{\frac{B}{N}\log\frac{2}{\delta_N}}$$

with probability at most $\delta_N/2$. Again by Hoeffding's inequality, for all points $\mu' \in \mathsf{U}^N$ we have

$$\frac{1}{N}\sum_{i=1}^{N}\mathbb{1}\left\{\mu_{(i)} \sim \mathbb{P}_\alpha \text{ and } \mu_{(i)} \in \mathbb{B}(\mu', r_{\mu'}) \setminus \mathbb{C}(\mu')\right\} \geq \alpha + \sqrt{\frac{B}{N}\log\frac{2}{\delta_N}}$$

with probability at most $\delta_N/2$, as $\mathbb{P}_\alpha\{\mu_{(i)} \in \mathbb{B}(\mu', r_{\mu'}) \setminus \mathbb{C}(\mu')\} \leq \alpha$ by the given condition. Therefore by definition, it follows that with probability at least $1 - \delta_N$ the distance $d_2$ satisfies $\left(\alpha + \sqrt{\frac{B}{N}\log\frac{2}{\delta_N}}, \nu + \sqrt{\frac{B}{N}\log\frac{2}{\delta_N}}\right)$-good neighborhood property. $\square$

### Proof of Theorem 3.2

*Proof.* From Lemma B.6, the distance $d_2$ satisfies $(\alpha', \nu')$-good property for the clustering problem $(\mathsf{U}^N, l)$. So there exists a subset $\mathsf{U}' \subset \mathsf{U}^N$ of size $(1 - \nu')N$ such that for any point $\mu' \in \mathsf{U}'$ all but $\alpha N$ out of $n_{\mathbb{C}(\mu') \cap \mathsf{U}'}$ neighbors in $\mathsf{U}'$ belongs to the cluster $C(\mu')$. For each $\mu' \in \mathsf{U}'$, let $r_{\mathsf{U}', \mu'}$ be the distance to the $n_{C(\mu') \cap \mathsf{U}'}$-th nearest neighbor of $\mu'$ in $\mathsf{U}'$, i.e.,

$$r_{\mathsf{U}', \mu'} := \inf\left\{r \geq 0 : |\mathsf{U}' \cap \mathbb{B}(\mu', r)| \geq n_{C(\mu') \cap \mathsf{U}'}\right\}. \tag{4}$$

Then it follows

$$|\mathsf{U}' \cap \mathbb{B}(\mu', r_{\mathsf{U}', \mu'}) \backslash \mathbb{C}(\mu')| \leq \alpha' N.$$

Now letting $\gamma = \sum_{a \in \mathcal{A}} \|\widehat{\mu}_a - \mu_a\|_\infty$, we define $\varepsilon$ as

$$\varepsilon := \sup_{\mu' \in \mathsf{U}'} \frac{|\mathsf{U}' \cap (\mathbb{B}(\mu', r_{\mathsf{U}', \mu'} + 4\gamma) \backslash \mathbb{B}(\mu', r_{\mathsf{U}', \mu'}))|}{N}.$$

Then by Corollary B.5, under Assumption A3, it follows that with probability $1 - \delta_N$,

$$\varepsilon \leq \sup_{\mu' \in \mathbb{R}^q, r > 0} \frac{|\mathsf{U}' \cap (\mathbb{B}(\mu', r + 4\gamma) \backslash \mathbb{B}(\mu', r))|}{N}$$

$$\leq 4\mathsf{C}_3 \left(\gamma + \frac{1}{N}\log\left(\frac{1}{\delta_N}\right) + \sqrt{\frac{\gamma}{N}\log\left(\frac{1}{\gamma}\right)}\right).$$

Hence,

$$\varepsilon = O\left(\gamma + \frac{1}{N}\log\left(\frac{1}{\delta_N}\right) + \sqrt{\frac{\gamma}{N}\log\left(\frac{1}{\gamma}\right)}\right).$$

Now we consider estimates of $\mathsf{U}^N$ (and correspondingly $\mathsf{U}'$). For each $\mu' \in \mathsf{U}^N$, let $\hat{\mu}'$ be an estimate of $\mu'$, and let $\widehat{\mathsf{U}}^N := \{\hat{\mu}' : \mu' \in \mathsf{U}^N\}$, and correspondingly, $\widehat{\mathsf{U}}' = \{\hat{\mu}' : \mu' \in \mathsf{U}'\} \subset \widehat{\mathsf{U}}^N$. On $\widehat{\mathsf{U}}^N$, define a cluster label $\hat{l} : \widehat{\mathsf{U}}^N \to \{C_1, \ldots, C_k\}$ as

$$\hat{l}(\hat{\mu}') = l(\mu'),$$

i.e., the cluster label $\hat{l}$ on $\hat{\mu}'$ coincides with the true cluster label $l$ on $\mu'$. Let $\hat{C}(\hat{\mu}')$ denote a cluster corresponding to $\hat{l}(\hat{\mu}')$, and define $\hat{\mathbb{C}}(\hat{\mu}') := \{\hat{\mu} : \hat{C}(\hat{\mu}) = \hat{C}(\hat{\mu}')\}$ as the set of $\hat{\mu}$ values for which $\hat{l}(\hat{\mu})$ matches $\hat{C}(\hat{\mu}')$. Then we have

$$n_{C(\mu') \cap \mathsf{U}'} = n_{\hat{C}(\hat{\mu}') \cap \widehat{\mathsf{U}}'}.$$

Now, note that $d_2(\mu, \mu') \leq r_{\mathsf{U}',\mu'}$ implies $d_2(\hat{\mu}, \hat{\mu}') \leq r_{\mathsf{U}',\mu'} + 2\gamma$, and hence $\mu \in \mathsf{U}' \cap \mathbb{B}(\mu', r_{\mathsf{U}',\mu'})$ implies $\hat{\mu} \in \widehat{\mathsf{U}}' \cap \mathbb{B}(\hat{\mu}', r_{\mathsf{U}',\mu'} + 2\gamma)$. Thus, it follows that

$$\left|\widehat{\mathsf{U}}' \cap \mathbb{B}(\hat{\mu}', r_{\mathsf{U}',\mu'} + 2\gamma)\right| \geq |\mathsf{U}' \cap \mathbb{B}(\mu', r_{\mathsf{U}',\mu'})| \geq n_{C(\mu') \cap \mathsf{U}'} = n_{\hat{C}(\hat{\mu}') \cap \widehat{\mathsf{U}}'}. \tag{5}$$

Therefore, if we define $\hat{r}_{\widehat{\mathsf{U}}',\hat{\mu}'}$ as the distance to the $n_{\hat{C}(\hat{\mu}') \cap \widehat{\mathsf{U}}'}$-th nearest neighbor of $\hat{\mu}'$ in $\widehat{\mathsf{U}}'$, similar to (4), as

$$\hat{r}_{\widehat{\mathsf{U}}',\hat{\mu}'} := \inf\left\{r \geq 0 : \left|\widehat{\mathsf{U}}' \cap \mathbb{B}(\hat{\mu}', r)\right| \geq n_{\hat{C}(\hat{\mu}') \cap \widehat{\mathsf{U}}'}\right\},$$

then, from (5), $\hat{r}_{\widehat{\mathsf{U}}',\hat{\mu}'}$ is bounded by

$$\hat{r}_{\widehat{\mathsf{U}}',\hat{\mu}'} \leq r_{\mathsf{U}',\mu'} + 2\gamma.$$

Also, note that $d_2(\hat{\mu}, \hat{\mu}') \leq r_{\mathsf{U}',\mu'} + 2\gamma$ implies $d_2(\mu, \mu') \leq r_{\mathsf{U}',\mu'} + 4\gamma$, and thereby $\hat{\mu} \in \widehat{\mathsf{U}}' \cap \mathbb{B}(\hat{\mu}', r_{\mathsf{U}',\mu'} + 2\gamma)$ implies $\mu \in \mathsf{U}' \cap \mathbb{B}(\mu', r_{\mathsf{U}',\mu'} + 4\gamma)$. Thus we have

$$\left|\widehat{\mathsf{U}}' \cap \mathbb{B}(\hat{\mu}', r_{\mathsf{U}',\mu'} + 2\gamma) \backslash \hat{\mathbb{C}}(\hat{\mu}')\right|$$
$$\leq |\mathsf{U}' \cap \mathbb{B}(\mu', r_{\mathsf{U}',\mu'} + 4\gamma) \backslash \mathbb{C}(\mu')|$$
$$\leq |\mathsf{U}' \cap \mathbb{B}(\mu', r_{\mathsf{U}',\mu'}) \backslash \mathbb{C}(\mu')| + |\mathsf{U}' \cap (\mathbb{B}(\mu', r_{\mathsf{U}',\mu'} + 4\gamma) \backslash \mathbb{B}(\mu', r_{\mathsf{U}',\mu'}))|$$
$$\leq (\alpha' + \varepsilon)N,$$

which leads to

$$\left|\widehat{\mathsf{U}}' \cap B(\hat{\mu}', \hat{r}_{\widehat{\mathsf{U}}',\hat{\mu}'}) \backslash \hat{\mathbb{C}}(\hat{\mu}')\right| \leq \left|\widehat{\mathsf{U}}' \cap \mathbb{B}(\hat{\mu}', r_{\mathsf{U}',\mu'} + 2\gamma) \backslash \hat{\mathbb{C}}(\hat{\mu}')\right| \leq (\alpha' + \varepsilon)N.$$

Consequently, the distance $d_2$ satisfies $(\alpha' + \varepsilon, \nu')$-good property for the clustering problem $(\widehat{\mathsf{U}}, \hat{l})$. Then as long as the smallest target cluster has size greater than $12(\nu' + \alpha' + \varepsilon)N$, Theorem A.1 implies that Algorithm 2 of Balcan et al. [5] with $n = \Theta\left(\frac{1}{\min(\alpha' + \varepsilon, \nu')} \log\left(\frac{1}{\delta \min(\alpha' + \varepsilon, \nu')}\right)\right)$ produces a hierarchy with a pruning that is $(\nu' + \delta)$-close to the target clustering with probability $1 - \delta - 2\delta_N$. $\quad\square$

**Proof of Theorem 4.1**

First, we give the following new result on bounding the Hausdorff distance between sets in the counterfactual function space.

**Theorem B.7.** *Suppose that $L_{h,t}$ is stable and let $H(\cdot, \cdot)$ be the Hausdorff distance between two sets. Suppose that Assumptions A1, A2, A3′, and A4 hold. Let $\delta \in (0, 1)$ and $\{h_n\}_{n \in \mathbb{N}} \subset (0, h_0)$ be satisfying*

$$\limsup_n \frac{(\log(1/h_n))_+ + \log(2/\delta)}{nh_n^q} < \infty.$$

*Then, with probability at least $1 - \delta$,*

$$H(\widehat{L}_{h_n,t}, L_{h_n,t}) \leq \mathsf{C}_{P,K,B}\left(\sqrt{\frac{(\log(1/h_n))_+ + \log(2/\delta)}{nh_n^q}}\right.$$

$$\left. + \frac{1}{h_n^{q+1}} \min\left\{\sum_a \|\widehat{\mu}_a - \mu_a\|_1 + \sqrt{\frac{\log(2/\delta)}{n}}, h_n\right\}\right)$$

In order to show Theorem B.7, we need the following Lemma.

**Lemma B.8.** *Suppose Assumptions A1, A2, A3′, and A4 hold. Let $\delta \in (0, 1)$ and $\{h_n\}_{n \in \mathbb{N}} \subset (0, h_0)$ be satisfying*

$$\limsup_n \frac{(\log(1/h_n))_+ + \log(2/\delta)}{nh_n^q} < \infty.$$

*Then, with probability at least $1 - \delta$,*

$$\|\widehat{p}_{h_n} - p_{h_n}\|_\infty$$

$$\leq \mathsf{C}_{P,K,B}\left(\sqrt{\frac{(\log(1/h_n))_+ + \log(2/\delta)}{nh_n^q}} + \frac{1}{h_n^{q+1}} \min\left\{\sum_a \|\widehat{\mu}_a - \mu_a\|_1 + \sqrt{\frac{\log(2/\delta)}{n}}, h_n\right\}\right).$$

*for some constant $\mathsf{C}_{P,K,B}$ depending only on $P$, $K$, $B$.*

For showing Lemma (B.8), we note that $\|\widehat{p}_{h_n} - p_{h_n}\|_\infty$ can be upper bounded as

$$\|\widehat{p}_{h_n} - p_{h_n}\|_\infty \le \|\widetilde{p}_{h_n} - p_{h_n}\|_\infty + \|\widehat{p}_{h_n} - \widetilde{p}_{h_n}\|_\infty. \tag{6}$$

Therefore, in what follows we shall provide high probability bound for $\|\widetilde{p}_{h_n} - p_{h_n}\|_\infty$ in Lemma (B.9) and $\|\widehat{p}_{h_n} - \widetilde{p}_{h_n}\|_\infty$ in Lemma (B.10). Then applying these to (6) will conclude the proof.

The following is from applying Kim et al. [37, Corollary 13].

**Lemma B.9.** *Under Assumptions A1, A2, A3′, and A4, if we let $\delta \in (0,1)$ and $\{h_n\}_{n \in \mathbb{N}} \subset (0, h_0)$ be satisfying*

$$\limsup_n \frac{(\log(1/h_n))_+ + \log(2/\delta)}{n h_n^q} < \infty,$$

*then with probability at least $1 - \delta$ it follows*

$$\|\widetilde{p}_{h_n} - p_{h_n}\|_\infty \le \mathsf{C}_{P,K} \sqrt{\frac{(\log(1/h_n))_+ + \log(2/\delta)}{n h_n^q}},$$

*where $\mathsf{C}_{P,K}$ depends only on $P$ and $K$.*

*Proof.* Consider $\mathbb{X} = \mathbb{B}(0, B + h_0)$. Then by Assumption (A2) for $\forall w \in \mathbb{R}^q \setminus \mathbb{X}$ it follows

$$\frac{\|\mu_{(i)} - w\|_2}{h} > 1.$$

$\operatorname{supp}(K) \subset \overline{B(0,1)}$ from Assumption (A4) implies that

$$\widetilde{p}_{h_n}(w) = \frac{1}{n} \sum_{i=1}^n K\left(\frac{\mu_{(i)} - w}{h}\right) = 0 \text{ a.s.},$$

and consequently that $p_{h_n}(w) = 0$ as well. Therefore,

$$\|\widetilde{p}_{h_n} - p_{h_n}\|_\infty = \sup_{w \in \mathbb{X}} |\widetilde{p}_{h_n}(w) - p_{h_n}(w)|. \tag{7}$$

Under Assumption A3′, $P$ has a bounded density $p$, so by Kim et al. [37, Proposition 5] we have that

$$\limsup_{r \to 0} \sup_{x \in \mathbb{X}} \frac{\int_{\mathbb{B}(x,r)} p(w) dw}{r^q} < \infty.$$

Note that under Assumption (A4), we have that $|K(x) - K(y)| \le M_K \|x - y\|_2$ for any $x, y \in \mathbb{R}^q$ and $\operatorname{supp}(K) \subset \overline{\mathbb{B}(0,1)}$, which together implies that $\|K\|_\infty \le M_K < \infty$. Hence,

$$\int_0^\infty t \sup_{\|x\| \ge t} K^2(x) dt \le \int_0^1 t M_K^2 dt = \frac{1}{2} M_K^2 < \infty.$$

Then applying Kim et al. [37, Corollary 13] gives that with probability at least $1 - \delta$,

$$\sup_{w \in \mathbb{X}} |\widetilde{p}_{h_n}(w) - p_{h_n}(w)| \le \mathsf{C}_{P,K} \sqrt{\frac{(\log(1/h_n))_+ + \log(2/\delta)}{n h_n^q}}, \tag{8}$$

where $\mathsf{C}_{P,K}$ depends only on $P$ and $K$. Finally (7) and (8) together imply that with probability at least $1 - \delta$,

$$\|\widetilde{p}_{h_n} - p_{h_n}\|_\infty \le \mathsf{C}_{P,K} \sqrt{\frac{(\log(1/h_n))_+ + \log(2/\delta)}{n h_n^q}}.$$

$\square$

**Lemma B.10.** *Under Assumptions A1, A2, and A4, Then*

$$\|\widehat{p}_{h_n} - \widetilde{p}_{h_n}\|_\infty \le \frac{\mathsf{C}_{M_K, B}}{h_n^{q+1}} \min\left\{\sum_a \mathbb{P}\left[\|\widehat{\mu}_a - \mu_a\|_1\right] + \sqrt{\frac{\log(1/\delta)}{n}}, h_n\right\},$$

*where $\mathsf{C}_{M_K, B}$ depends only on $M_K$ and $B$.*

*Proof.* By Assumption A4 it follows that $|K(x) - K(y)| \leq M_K \|x - y\|_2$ for any $x, y \in \mathbb{R}^q$ and $\operatorname{supp}(K) \subset \overline{\mathbb{B}(0,1)}$, which together implies that $|K(x) - K(y)| \leq M_K$ and $\|K\|_\infty \leq M_K$. Thus it follows

$$|K(x) - K(y)| \leq \min\{M_K \|x - y\|_2, M_K\}.$$

Now for any $w \in \mathbb{R}^q$, $|\widehat{p}_{h_n}(w) - \widetilde{p}_{h_n}(w)|$ is upper bounded by

$$
\begin{aligned}
|\widehat{p}_{h_n}(w) - \widetilde{p}_{h_n}(w)| &\leq \frac{1}{nh_n^q} \sum_{i=1}^n \left| K\left(\frac{\widehat{\mu}_{(i)} - w}{h_n}\right) - K\left(\frac{\mu_{(i)} - w}{h_n}\right) \right| \\
&\leq \frac{1}{nh_n^q} \sum_{i=1}^n \min\left\{ M_K \frac{\|\widehat{\mu}_{(i)} - \mu_{(i)}\|_2}{h_n}, M_K \right\} \\
&\leq \frac{M_K}{h_n^{q+1}} \min\left\{ \frac{1}{n} \sum_{i=1}^n \|\widehat{\mu}_{(i)} - \mu_{(i)}\|_2, h_n \right\}.
\end{aligned}
$$

Since this holds for any $w \in \mathbb{R}^q$,

$$\|\widehat{p}_{h_n} - \widetilde{p}_{h_n}\|_\infty \leq \frac{M_K}{h_n^{q+1}} \min\left\{ \frac{1}{n} \sum_{i=1}^n \|\widehat{\mu}_{(i)} - \mu_{(i)}\|_2, h_n \right\}.$$

Then under (A4), applying (3) from Lemma B.1 gives that with probability $1 - \delta$, $\|\widehat{p}_{h_n} - \widetilde{p}_{h_n}\|_\infty$ is upper bounded as

$$
\begin{aligned}
\|\widehat{p}_{h_n} - \widetilde{p}_{h_n}\|_\infty &\leq \frac{M_K}{h_n^{q+1}} \min\left\{ \|\widehat{\mu}_a - \mu_a\|_1 + 2B\sqrt{\frac{\log(1/\delta)}{n}}, h_n \right\} \\
&\leq \frac{\mathsf{C}_{M_K, B}}{h_n^{q+1}} \min\left\{ \|\widehat{\mu}_a - \mu_a\|_1 + \sqrt{\frac{\log(1/\delta)}{n}}, h_n \right\},
\end{aligned}
$$

where $\mathsf{C}_{M_K, B} = M_K \max\{1, 2B\}$.

$\square$

Now we are ready to prove Lemma B.8.

*Proof of Lemma B.8.* As in (6), we upper bound $\|\widehat{p}_{h_n} - p_{h_n}\|_\infty$ as

$$\|\widehat{p}_{h_n} - p_{h_n}\|_\infty \leq \|\widehat{p}_{h_n} - \widetilde{p}_{h_n}\|_\infty + \|\widetilde{p}_{h_n} - p_{h_n}\|_\infty.$$

Then by Lemma B.9 and B.10, with probability $1 - \delta$ it follows that

$$
\begin{aligned}
&\|\widehat{p}_{h_n} - p_{h_n}\|_\infty \\
&\leq \mathsf{C}_{P,K} \sqrt{\frac{(\log(1/h_n))_+ + \log(2/\delta)}{nh_n^q}} + \frac{\mathsf{C}_{M_K, B}}{h_n^{q+1}} \min\left\{ \sum_a \|\widehat{\mu}_a - \mu_a\|_1 + \sqrt{\frac{\log(2/\delta)}{n}}, h_n \right\} \\
&\leq \mathsf{C}_{P,K,B} \left( \sqrt{\frac{(\log(1/h_n))_+ + \log(2/\delta)}{nh_n^q}} + \frac{1}{h_n^{q+1}} \min\left\{ \sum_a \|\widehat{\mu}_a - \mu_a\|_1 + \sqrt{\frac{\log(2/\delta)}{n}}, h_n \right\} \right),
\end{aligned}
$$

where $\mathsf{C}_{P,K,B}$ depends only on $P, K, B$.

$\square$

We are now in a position to prove Theorem 4.1.

**Proof of Theorem 4.1**

Recall that $L_{h_n,t}$ is stable if there exist $a > 0$ and $C > 0$ such that, for all $0 < \zeta < a$, $H(L_{h_n,t-\zeta}, L_{h_n,t+\zeta}) \leq C\zeta$.

*Proof.* Let us define

$$r_n := \mathsf{C}_{P,K,B}\left(\sqrt{\frac{(\log(1/h_n))_+ + \log(2/\delta)}{nh_n^q}} + \frac{1}{h_n^{q+1}}\min\left\{\sum_a \|\widehat{\mu}_a - \mu_a\|_1 + \sqrt{\frac{\log(2/\delta)}{n}}, h_n\right\}\right),$$

which is RHS of the inequality in Lemma B.8.

Suppose that we are given a sufficiently large $n$ so that $\|\widehat{p}_{h_n} - p_{h_n}\|_\infty < r_n$ holds with probability at least $1 - \delta$ where $r_n < a$ for some constant $a > 0$. We aim to show two things: (a) for every $x \in L_{h_n,t}$ there exists $y \in \widehat{L}_{h_n,t}$ with $\|x - y\|_2 \leq Cr_n$, and (b) for every $x \in \widehat{L}_{h_n,t}$ there exists $y \in L_{h_n,t}$ with $\|x - y\|_2 \leq Cr_n$.

To show (a), consider $x \in L_{h_n,t}$, Then by the stability property of $L_{h_n,t}$, there exists $y \in L_{h_n,t+r_n}$ such that $\|x - y\|_2 \leq Cr_n$. Then $p_{h_n}(y) > t + r_n$ which implies that

$$\widehat{p}_{h_n}(y) \geq p_{h_n}(y) - \|\widehat{p}_{h_n} - p_{h_n}\|_\infty > p_{h_n}(y) - r_n > t.$$

Hence we conclude $y \in \widehat{L}_{h_n,t}$ with $\|x - y\|_2 \leq Cr_n$.

Similarly, to show (b), consider $x \in \widehat{L}_{h_n,t}$ so that $\widehat{p}_{h_n}(x) > t$. Thus we have

$$p_{h_n}(x) \geq \widehat{p}_{h_n}(x) - \|\widehat{p}_{h_n} - p_{h_n}\|_\infty > t - r_n,$$

which leads to $x \in L_{h_n,t-r_n}$. Then again by the stability property of $L_{h_n,t}$, there exists $y \in L_{h_n,t}$ such that $\|x - y\|_2 \leq Cr_n$.

Hence by definition, we upper bound the Hausdorff distance $H(\widehat{L}_{h,t}, L_{h,t})$ by

$$Cr_n$$

$$= C\mathsf{C}_{P,K,B}\left(\sqrt{\frac{(\log(1/h_n))_+ + \log(2/\delta)}{nh_n^q}} + \frac{1}{h_n^{q+1}}\min\left\{\sum_a \|\widehat{\mu}_a - \mu_a\|_1 + \sqrt{\frac{\log(2/\delta)}{n}}, h_n\right\}\right).$$

$\square$

