# OpenReview forum: "Hierarchical and Density-based Causal Clustering"
_NeurIPS.cc/2024/Conference — NeurIPS 2024 poster_

### Official Review · Reviewer_xeku · 2024-07-11

**Soundness:** 3
**Presentation:** 3
**Contribution:** 3
**Rating:** 6
**Confidence:** 2

**Summary:**

The paper aims to understand treatment effect heterogeneity and identify and evaluate subgroup effects. It addresses the challenge of typically unknown subgroup structures by proposing a solution based on causal k-means clustering to assess effect heterogeneity. The approach is improved by integrating hierarchical and density-based clustering algorithms, providing a more nuanced and effective method for identifying and evaluating subgroup effects.

**Strengths:**

The paper exhibits several strengths across originality, quality, clarity, and significance:

Originality: The introduction of simple plug-in estimators, implementable with off-the-shelf algorithms, is notable. This approach opens new avenues for clustering with generic pseudo-outcomes and contributes to identifying homogeneous subgroups in treatment response.
Quality: The paper thoroughly explores finite sample properties via simulation, with a setup similar to prior studies. The rate of convergence is presented, and the inclusion of standard error bars in the graphs adds to the reliability of the results.
Clarity: The experiments are straightforward and concise, with all necessary details provided. The graphs are clear and effectively convey the results.
Significance: The paper significantly contributes to the progression of methodologies for identifying homogeneous subgroups in treatment response.

**Weaknesses:**

The paper lacks a comparative analysis of the method's performance against existing methods in the empirical analysis section. Specifically, it does not demonstrate if the new method captures the underlying structure more accurately than other existing methods. Including such comparisons would strengthen the paper and make its contributions more convincing.

**Questions:**

1. Among the case studies, the number of covariates is very small. How does the convergence rate of \hat{\mu}_a change as the dimension of the covariate space increases? Specifically, how slow does it get in practice as the number of covariates grows?
2. Do the authors have suggestions on the optimal number of covariates to limit to before applying this method in practice?
3. Are there any methods the authors can recommend to help identify important covariates that modify the treatment effect?

**Limitations:**

The authors have addressed the limitations in their discussion section, acknowledging that this method is a useful tool for exploring subgroup structures. They recommend using other methods in combination with their proposed method to inform specific decisions.

One area for improvement is the interpretation of the subgroup cluster results, which poses another challenge. Could the authors explain how to better interpret the resulting clusters in their application example? Additionally, how can this information be utilized to inform decisions afterward? Providing more guidance on these aspects would enhance the practical applicability of their work.

---

> ### Author Rebuttal · Authors · 2024-08-06
>
> We appreciate your valuable comments and suggestions. We address each of
> them below.
>
> 1.  **\[Experimental comparison\]** Yes, we completely agree that our
>     work could be strengthened by experimentally comparing with other
>     clustering methods, especially if other methods require any special
>     assumptions that are not present in ours (like the margin-type
>     conditions as in k-means) to be employed in the causal clustering
>     framework. However, this would require a closer analysis at those
>     methods. If other alternatives do not require particularly stronger
>     assumptions and can be readily integrated into the causal clustering
>     framework as well (though this must be theoretically verified as in
>     our work), then the problem turns to which approach to use for
>     specific data. Because each clustering algorithm has benefits and
>     drawbacks, it is difficult to conduct a fair comparison using
>     experiments. Given these circumstances, we believe that designing a
>     fair experiment to demonstrate the superiority of our proposed
>     methods is not quite straightforward, and that's outside the scope
>     of this paper.
>
>     Furthermore, comparisons with other \"causal clustering\"
>     counterparts in the SCM literature appear unclear because they are
>     designed to analyze structural heterogeneity, whereas ours is
>     designed to analyze treatment effect heterogeneity. We may consider
>     some special data generation process where we may demonstrate
>     superiority of our method, yet this might not be convincing.
>
>     All in all, we believe that confining our work to presenting the
>     respective theory that the two appealing off-the-shelf
>     cluster-analysis methods can be successfully adopted within the
>     novel framework of causal clustering would suffice for the time
>     being. Nonetheless, following your comment, we will go over the
>     potential extension to other clustering algorithms in the discussion
>     section in the revised manuscript.
>
> 2.  **\[Ans to Q1\]** This is a good question. The rate of convergence
>     of regression functions in nonparametric modeling is well studied
>     (e.g., Györfi, 2002). It depends on both the function space to which
>     the true $\mu_a$ belongs and the estimator itself $\hat{\mu}_a$;
>     e.g., for $\mu_a$ in the Holder class with smoothness $s$, given
>     $dim(X)=d$, the best (in minimax sense) rates are
>     $O(n^{-\frac{s}{2s+d}})$. If $\mu_a$ is not smooth enough, then
>     high-dimensional covariates will increase uncertainty of our
>     estimators.
>
>     In fact, in our simulation in Section 5.1., we incorporated the
>     effect of the number of covariates by directly controlling the
>     convergence rate through a parameter $\beta$, i.e., by letting
>     $\Vert \hat{\mu}_a - \mu_a\Vert = O_P(n^{-\beta})$. The results are
>     presented in Figure 2 (we will magnify this figure in the revised
>     manuscript).
>
> 3.  **\[Ans to Q2\]** In our opinion, there is no precise answer to this
>     question. As previously stated, the smaller the number of covariates
>     ($=d$), the lower the estimation error. However, additional
>     components also interact with $d$. For example, in density-based
>     clustering, we have bandwidth that is entangled with $d$, both of
>     which affect the error. More importantly, if we use only a small
>     number of covariates, the no-unmeasured confounding assumption
>     (Assumption C2) is likely to be violated; we usually collect as many
>     covariates as possible to ensure there is no nmeasured confounders
>     left. All of these factors must be considered simultaneously.
>
> 4.  **\[Ans to Q3\]** This leads to an entirely different, but
>     interesting question. Given a collection of covariates, one can
>     accurately estimate the CATE function using approaches such as
>     Kennedy (2023), and then apply an algorithm to determine which
>     covariate (or combination of covariates) has the most impact on the
>     treatment effect. Or based on our methods, one can analyze the
>     covariate distributions across clusters and determine which
>     covariate results in the greatest distributional divergence in
>     subgroup effects. We will add this comment in our revised
>     manuscript.
>
> 5.  **\[Interpretation\]** This is also a great question. One possible
>     interpretation is that each cluster has its own subpopulation from
>     which units are generated. Properties of each subpopulation could be
>     studied by analyzing distributional features, etc. However, as we
>     highlighted in the Discussion section as a shortcoming, while our
>     approach allows for efficient discovery of subgroup structures,, it
>     may be less effective for prescriptive applications (i.e., informing
>     specific treatment decisions). Thus, it is important to exercise
>     caution while attempting to interpret the observed clusters.

---

### Official Review · Reviewer_Ykrk · 2024-07-12

**Soundness:** 4
**Presentation:** 3
**Contribution:** 3
**Rating:** 6
**Confidence:** 3

**Summary:**

This work solves the task of clustering treatment-effect data based on their conditional treatment effect with a discrete set of treatments and continuous (possibly multivariate) effects. More specifically, they extend the framework of previous work on causal k-means which achieves the same, albeit which is now applied on density-based (hierarchical) clustering.

Similar to the work on causal k-means, the authors opt for clustering samples based on their signature in the CATE space, i.e., a Euclidean vector whose elements are the avg. treatment effect given each treatment, jointly with an appropriate plugin estimator. To bound the error of the resulting pruning tree, they first extend the concept of $(\alpha,\nu)-$good neighbourhood to incorporate a general distribution (in the original work the empirical one is is used) and then adapt the main result of that work to their extension. To extend this framework to hierarchical clustering they employ Balcan et al's (2014) method, which provides an algorithm that has low (up to O(n) ) complexity and is robust to outliers, among other traits. Importantly, they extend the error bounds of the latter method to the causal case, which shows a small discounting of the accuracy due to the added CATE estimation overhead.

They demonstrate their result on synthetic and real world datasets, which are nicely visualised for 2D.

**Strengths:**

1. The work provides theorems to rigorously derive a bound on the error of the resulting outcome.
2. The assumptions make sense and are well studied.

**Weaknesses:**

# Contextualisation
This work does mention several other methods closel or loosely related with the task at hand, however none of these other methods have been compared against in the experiments. Although this could well be justified as providing an isolated work providing the respective theory for this specific approach, the case that the authors make could be strengthened by comparing with other methods. For instance, those extracting causal rules, or a step-wise clustering approach that first estimates the CATE and then applies clustering on it. Other density based clustering methods could also be used, for instance hierarchical DBSCAN [1] which is also considered a robust method, or even with causal k-means on the estimated CATE vectors. The point here would not necessarilly be to justify the superiority of Balcan et al's method (which arguably is also not too concerned with this in the first place), but to also study experimentally the superiority of their theoretically superior results.

```
[1] Ricardo J. G. B. Campello, Davoud Moulavi, Joerg Sander: Density-Based Clustering Based on Hierarchical Density Estimates. In: Advances in Knowledge Discovery and Data Mining. Springer Berlin Heidelberg, Berlin, Heidelberg 2013
```

Additionally, this work borrows from two key works, the causal k-means and the Balcan et al's method, but the extensions and modifications of thie present work are contributions which can at some times be unclear.

# Plug in estimator
The kernel density estimate can often be criticised due to its weaknesses, for instance its susceptibility to the curse of dimensionality, need for hyperparameter tuning, etc. How does the method depend on the choice of kernel, beside its Lipschitz constant?

# Presentation
First, I was bothered by the $\equiv$ sign in line 180. This doesn't seem to be a standard or previously introduced notation. What is the advantage of using this over an equals sign? Do you want to stress that $\mathbb C$ is some ball, itself?

Additionally, I believe it would be easier for the reader if you broke down the definition 3.1 of or provided intuitive explanations of the $\nu$-strict and $\alpha$-good parts of it, similar to the Balcan et al. work, mutatis mutandis for your distirbution-general extension.

Some minor typos:
* 115: us harness$\rightarrow$us to harness
* 200: have$\rightarrow$has
* 210: arbitrarily sized
* The legend and labels in Figure 4 are too small to be legible when printed on paper.

**Questions:**

1. In your work you make an extensive ue of the density of the sample points $(X,Y)$; however, the Housdorff distance definition you provide (ln. 247) seems to be oblivious to this distribution, only sensitive to the envelope of these points. Could you comment on this?

2. See questions on presentation above.

3. You may comment on comparisons section, above.

4. You may comment on the plug in estimator as mentioned above.

5. The use of CATE as a clustering domain seems to have its merits; here, the conditioning set seems to be completely ignored, up to the estimation of these vectors, themselves. Say, two very different patients (in terms of X) that have similar CATE profiles would be clustered together; from a standard causal perpsective using a DAG-based SCM, this would amount to splitting the dataset in sub-parts (clusters) which might not be well aligned with the varialbes of the underlying SCM, and could make downstream structural causal learning tasks harder. Could you comment on this?

6. Could you comment on my understanding of the hyperparameter need (see limitations)?

**Limitations:**

The authors make a good effort to be open on the criticisms of certain standard assumptions in the field of causality, which do not, however, hurt the validity of the method.

Another basic limitation that seems to be inherited from the use of Balcan et al's work which requires the a-priori specification of the $\alpha, \nu$ hyper parameters, for which there does not seem to be a good way to intuitively specify.

---

> ### Author Rebuttal · Authors · 2024-08-06
>
> We appreciate your valuable input and insights. We address each of your
> comments below.
>
> 1.  **\[Clarification on contributions\]** Thank you for bringing our
>     attention to this. The main contributions of our work is that we
>     have proved that the two appealing off-the-shelf cluster-analysis
>     techniques can be successfully adopted within the novel framework of
>     causal clustering, using simple plug-in methods without requiring
>     additional strong structural assumptions as opposed to k-means,
>     which requires the margin condition (kim et al. 2024). For example,
>     in Section 3, we demonstrate that the robust inductive hierarchical
>     clustering method (Balcan el al. 2014) may be applied to causal
>     clustering through the plug-in estimator with minimal assumptions,
>     and we analytically validate the associated costs in Theorem 3.1.
>     Verifying this is not as straightforward as one may imagine. We will
>     state our contributions more clearly in Section 1, particularly in
>     relation to kim et al. (2024) and key references in Sections 3 & 4.
>
> 2.  **\[Experimental comparison\]** Relatedly, yes, we completely agree
>     that our work could be strengthened by experimentally comparing with
>     other methods such as hdbscan, if other methods require any special
>     assumptions that are not present in ours (like the margin-type
>     conditions as in k-means) to be employed in the causal clustering
>     framework. However, this would require a closer analysis at those
>     methods. If other alternatives do not require particularly stronger
>     assumptions and can be readily integrated into the causal clustering
>     framework as well (though this must be verified), then the problem
>     turns to which approach to use for specific data. Since each
>     clustering algorithm has its own pros and demerits, it is difficult
>     to conduct a fair comparison using experiments. Given these
>     circumstances, we believe that designing a fair experiment to
>     demonstrate the superiority of our proposed methods is not quite
>     straightforward, and that's outside the scope of this paper. (Please
>     let us know if we misinterpret your intention here). Following your
>     comment, however, we will provide a brief discussion of the
>     potential extension to other clustering algorithms.
>
> 3.  **\[Plug-in KDE\]** As pointed out, our plug-in estimator
>     essentially inherits pros and cons of the standard KDE (in level-set
>     clustering). In theory and practice, the choice of the kernel
>     affects the performance of KDE through the bandwidth, and when the
>     bandwidth is appropriately chosen, the shape of the kernel has
>     little effect on the performance. In kernel density estimation
>     (KDE), Gaussian kernel has the advantage that the number of modes
>     monotonically decreases as the bandwidth increases (Silverman, 1981,
>     Using kernel density estimates to investigate multimodality). In
>     kernel regression, Epanechnikov kernel is the most efficient in the
>     constant term of mean integrated squared error, but other usual
>     kernels (including Gaussian) are at least \~90% efficient. Other
>     than these effects, the shape of the kernel does not affect the
>     convergence rate and has little effect on learning error by less
>     than \~10% on the constant. The choice of kernel has no effect on
>     our results (but it may in real data studies).
>
> 4.  **\[Issues with presentation\]** Thank you for pointing these out.
>
>     -   We shall remove the $\equiv$ signs, which had been often used to
>         emphasize notational \"equivalence,\" and replace them with
>         standard equal signs.
>
>     -   We totally agree it should be better to broke down the
>         definition 3.1 as in the original work of Balcan et al. Will do
>         that accordingly in the revised manuscript (or at least in the
>         appendix).
>
>     -   Thank you for correcting the typos; during revision, we will fix
>         everything and proofread thoroughly.
>
> 5.  **\[Ans to Q1\]** As you have mentioned, the Hausdorff distance
>     $H(S_{1}, S_{2})$ is oblivious to the density but only sensitive to
>     the envelope: given that we already have enough points in a specific region, adding more points does not meaningfully change the Hausdorff distance. However, we are looking at the level sets $L_{t,h}=\\{w\in \mathbb{R}^{q}: p_{h}(w)>t\\}$ for $t>0$, and their estimators $\hat{L}\_{t,h}$ for $t>0$. Then the density information is encoded through the data points in the level sets. For example,
>     if one distribution has high density regions but the other does not,
>     then for a large $t>0$, the level set
>     ${L}\_{t,h}$ for the first
>     distribution would be nonempty, while the level set for the second
>     distribution would be empty. Hence, although the Hausdorff distance
>     is oblivious to the density, the density is encoded through the
>     level sets $L\_{t,h}$, and measuring the difference by the Hausdorff
>     distance behaves sensitively with respect to the density of data
>     points.
>
> 7.  **\[Ans to Q5\]** We believe your insight is correct. We don't yet
>     have a good answer to how to reconcile with the related structural
>     causal learning tasks. As we leave comments for reviewer STix,
>     causal clustering using the SCM approach may yield different results
>     from ours. We will at least comment on this in the discussion
>     section, with the hope of opening up new options for subsequent
>     research.
>
> 8.  **\[Ans to Q6\]** You are correct - just like Balcan et al (2014)
>     our estimator requires tuning the two noise parameters. We do not
>     propose any good solution for parameter selection in our paper.
>     Nonetheless, Balcan et al. have empirically showed robustness to
>     such parameter tuning, thus we believe our method can inherit this
>     property, albeit further work is needed because these parameters
>     play an essential part in our method. We will add this discussion in
>     the revised work.

---

> > ### Comment · Reviewer_Ykrk · 2024-08-12
> >
> > I thank the reviewers for their effors in responding to my questions.
> >
> > Thank you for your responce to Q1, as it does address my question. I believe your answer to Q5 is something that could be considered as a limitation of the general approach of this method, which does still not seem to be addressed in this work. If accepted, I believe you should also make the limitation of your answer to Q6 explicit in your work.
> > I am also aligned with remarks of other reviewers on the extent of novelty.
> >
> > Overall, I will be maintaining my gently positive score, albeit with limited fervor.

---

> ### Author Response · Authors · 2024-08-14
> **Thank you**
>
> Thank you very much for your valuable feedback. As you suggested, we will add a new paragraph on Q5 and Q6 as the limitation of our work in the discussion section, hoping it opens a new avenue for future research. We are confident that incorporating your suggestions above will significantly enhance the quality of our paper.

---

### Official Review · Reviewer_STix · 2024-07-13

**Soundness:** 2
**Presentation:** 2
**Contribution:** 3
**Rating:** 7
**Confidence:** 3

**Summary:**

The authors propose an extension of existing causal (i.e., treatment effect heterogeneity) k-means clustering techniques to hierarchical and density-based clustering, including novel estimators and convergence guarantees.

**Edit**: increased rating from 3 to 7 (with the understanding that more qualified people (ACs/PCs/ethicists) will look into the plagiarism concern)

**Strengths:**

The topic of causal clustering has already been addressed from variety of perspectives in previous work (e.g., kernel methods suitable for k-means, hierarchical, and density-based clustering, as well as focusing on heterogeneity in terms of causal structure or causal mechanisms), but this work nevertheless finds an original approach (hierarchical/density-based clustering for treatment effect heterogeneity), which is clearly motivated, has high quality theoretical justifications, and should be significant in the (theoretical and applied) causal inference community.

**Weaknesses:**

The main weakness (and why my overall rating is a 3 instead of more like a 6 or 7), is the inappropriate verbatim copying from reference [36] (Kwangho Kim, Jisu Kim, and Edward H Kennedy. Causal k-means clustering. arXiv preprint arXiv:2405.03083, 2024.), including the following (and I assume more):
- lines 7,8: "We present..."
- lines 18--20: mostly copied verbatim from abstract of [36]
- lines 91--99: in its entirety
- lines 104,105: "If all coordinates..."

The figures (espcially 4, which requires over 400% maginification) are too small to be legible on standard paper sizes, making it harder to understand or corroborate the results described in the text. Furthermore, the unreasonably small figures save up to a page of space, allowing more text to be squeezed into the page limit, which seems unfair considering the submission guidelines.

I find the related work discussed in Section 1.2 to be lacking. I would expect to see some reference and discussion/comparison with other previous work on causal clustering (which has focused more on heterogeneity in terms of causal mechanisms and causal structure rather than treatment effect), for example including:
- Hu, S., Chen, Z., Partovi Nia, V., Chan, L., & Geng, Y. (2018). Causal inference and mechanism clustering of a mixture of additive noise models. Advances in Neural Information Processing Systems, 31.
- Huang, B., Zhang, K., Xie, P., Gong, M., Xing, E. P., & Glymour, C. (2019). Specific and shared causal relation modeling and mechanism-based clustering. Advances in Neural Information Processing Systems, 32.
- Saeed, B., Panigrahi, S., & Uhler, C. (2020). Causal structure discovery from distributions arising from mixtures of DAGs. In International Conference on Machine Learning (pp. 8336-8345). PMLR.
- Markham, A., Das, R., & Grosse-Wentrup, M. (2022). A distance covariance-based kernel for nonlinear causal clustering in heterogeneous populations. In Conference on Causal Learning and Reasoning (pp. 542-558). PMLR.

Maybe experimental comparison against some of the above methods is also possible/desirable?

**Questions:**

1. line 134: Is $d$ a distance function (which by definition is positive when evaluated on distinct objects) or some weaker notion with image $[-1, 1]$?
2. line 164: Can the authors elaborate on why "the true target hierarcy... is an infinite set of clusters"?
3. line 338: What are some examples of such subsequent learning tasks?

**Limitations:**

Assumptions are clearly stated throughout the text, and general limitations are explicitly discussed at the end.

However, I would expect to see some discussion about how this work (which facilitates targeted interventions on specific subgroups) relates to issues of fairness/bias. At the very least, a more complete answer to Question 10 in the author checklist should be given, following the guidelines: "If the authors answer NA or No, they should explain why their work has no societal impact".

---

> ### Author Rebuttal · Authors · 2024-08-06
>
> Thank you for your time and thorough feedback. We have addressed each of your concerns as outlined below.
>
> 1. **[Originality and related work]** Thank you for highlighting the connection between our research and other works in the literature on causal discovery or structural causal models. To our knowledge, there are three common ways to express causal/counterfactual quantities: (1) structural equations, (2) causal graphs or structural models, and (3) potential outcomes (counterfactuals). While these languages can complement each other, the target parameters, assumptions, notations, and techniques often differ. We acknowledge that the prior works you mentioned also address the notion of "causal clustering." However, they fall into the second approach (in the context of learning *structural heterogeneity*), whereas ours is based on the third (learning *treatment effect heterogeneity*). This distinction is why we did not include the extensive body of work in causal discovery/structural causal models. To the best of our knowledge, Kim et al. (2024) were the first to formally study the task of analyzing treatment effect heterogeneity via clustering based on the (potentially multivariate) CATE within the potential outcomes framework. In our revised manuscript, we will clearly highlight this difference and define our "causal clustering" problem. Additionally, based on your comments, we will add a separate subsection in Section 1 explaining the connection between our approach and others in the causal structure learning literature.
>
> 2. **[Figures and Presentation]** Thank you for pointing this out. We agree that some figures, particularly Figure 4, are too small. Since the essential idea to be presented is quite macroscopic (i.e., cluster patterns), we believe that expanding it slightly larger with increased font and legend size can alleviate this issue. We will address this in the amended text, and if necessary, we will move some figures to the Appendix.
>
> 3. **[Inappropriate verbatim from [36]]** We appreciate your suggestion regarding this matter. Our reliance on the previous work of Kim et al. (2024) is mainly for describing the motivation, problem, and setup in Sections 1 and 2. The framework is novel within the community, and our intention is to present the problem accurately without distorting its description. (As mentioned above, our problem differs significantly from the causal clustering framework in the causal structure learning literature.) However, we acknowledge that we should avoid using verbatim as much as possible and rephrase where necessary. We will fully address this in the revised text.
>
>     Nonetheless, we would like to emphasize that Sections 3 and 4, which constitute the main contribution of our paper, are entirely original and do not contain any issues related to verbatim content from [36]. Therefore, in our opinion, this should not significantly diminish the value of our paper.
>
> 4. **[Ans to Q1]** We apologize: it should be a distance with the image $[0,1]$. We will fix this.
>
> 5. **[Ans to Q2]** We apologize for the confusion. As you noted, it is an erroneous expression, and part of the sentence should be revised as follows: "... with respect to the true target clustering, because we build a set of nested clusters across various resolutions (a hierarchy) such that the target clustering is close to some pruning of that hierarchy."
>
> 6. **[Ans to Q3]** These could be utilized, for example, to develop precision medicine or optimal policy. We will add specific examples in the discussion section.
>
> 7. **[Ethics Review]** This is a very good point. In the last section, we will discuss how the discovered subgroup was formed simply based on similarity in treatment effect, without considering factors such as fairness/bias. We believe that discovering a "fair subgroup" would be an intriguing future project of our work.
>
> If there are any remaining issues, please feel free to let us know through your comments.

---

> > ### Comment · Reviewer_STix · 2024-08-10
> >
> > Thanks for the thorough rebuttal---it addresses all of my concerns! The other reviews and corresponding rebuttals also gives me a more favorable view of the paper. I have **increased my rating** from 3 to **7**, with the understanding that more qualified people (ACs/PCs/ethicists) will look into the plagiarism concern.
> >
> > In summary:
> > Causal clustering for treatment effect heterogeneity is a well-motivated problem of practical importance, and this paper adds to the (limited) existing literature in a natural direction, offering solid theoretical results and basic proof-of-concept empirical results. I would expect this paper to have high impact in the causal inference community (considering both the theoretical and practical sides).

---

> ### Author Response · Authors · 2024-08-10
> **Thank you**
>
> We appreciate your helpful questions and comments, and we feel that the revised manuscript will be significantly improved as a result. Most importantly, we appreciate the reviewer pointing out the issue of several inappropriate verbatims while we define the problem and setup in Section 2. We take this issue extremely seriously and will fully address it in the revised manuscript. Please feel free to let us know if you have any further concerns or questions.

---

### Official Review · Reviewer_geg3 · 2024-07-16

**Soundness:** 3
**Presentation:** 2
**Contribution:** 2
**Rating:** 3
**Confidence:** 2

**Summary:**

The paper deals with problems arising in understanding treatment response/effects and in particular evaluating subgroup effects building on recent work using causal k-means clustering. The main contribution of the paper is to circumvent the k-means approach and suggest a hierarchical and also a density-based clustering approach. The authors present associated estimators with their proposed methods and rate of convergence, thus extending the framework of causal clsutering.

The paper is motivated by the study of heterogeneous treatment effects via clustering and highlights the drawbacks of prior work based on k-means. The authors extend this by applying density-based clustering which has the advantage of finding clusters with arbitrary shapes and sizes and appears to be more robust to noise and outliers. Similarly, the observe that hierarchical clustering has some advantages in scenarios where the data are nested or are forming hierarchies.

The main results are:

-Th. 3.1: The authors analyze the robust hierarchical clustering algorithm appearing in prior works [5], in the context of causal clustering. Under certain assumptions, that are related to so-called good-neighborhood properties as defined in [5] the authors manage to show that having access to a small random subset of the data can allow for their algorithm to have small error on the entire data set.

-Th. 4.1: Analogous statement for the density-based clustering methods.

Finally the authors presented experiments for studying finite-sample properties of their proposed plug-in procedures using simulated data.

**Strengths:**

+analysis for robust hierarchical clustering and density-based methods seems interesting

+natural algorithms and problems are well-motivated

**Weaknesses:**

-despite the well-motiaveted setting, the reviewer believes that the paper has a lack of novelty: the algorithms are already from previous works, and most of the approaches are what was expected.

-presentation can be improved. As of now, it is a bit cryptic, as the main tasks/problems are not well-defined and rather are to be implied from the context. I would have loved to see a clean definition of the problem that is being solved, the challenges and the novel approach. As of now, I believe the results are a plug-in approach based on two well-studied algorithms.

**Questions:**

-To improve the results, I was wondering if there are other notions of error, for which your results can be extended? What if we measure error for example on the hierarchy that is being found? There are various notions for comparing hierarchies in the literature and I believe this could be interesting and novel extension of your work.

**Limitations:**

-see above.

---

> ### Author Rebuttal · Authors · 2024-08-06
>
> Thank you for your time and valuable feedback. We would like to address each of your major concerns in the following.
>
> 1. **[Novelty of the problem]** Thank you for pointing this out. First, we want to emphasize that the problem of causal clustering is novel; clustering on counterfactual outcomes provides a new framework for analyzing treatment effect heterogeneity, and as far as we know this approach has not addressed (or at the very least, formally addressed) in the literature before, except in Kim et al. (2024). It also differs significantly from other previous attempts in the cluster analysis literature that considered partially observed outcomes or clustering with measurement errors, since in our case the variable to be clustered consists of completely unknown counterfactual functionals (See Section 1.2 of Kim et al. (2024)). Due of the page limits, we just briefly describe the problem in Section 2 and refer readers to Kim et al. (2024) for details, which could confuse them about the problem's novelty. Based on the reviewer's comment, we will explicitly clarify on the problem, setting, and associated challenges.
>
> 2. **[Contribution]** We would like to stress out that our main contribution is not on the algorithm side, i.e., the use of the robust hierarchical or density-based clustering algorithms. Rather, it is on the theoretical side where we provide conditions under which the plug-in estimator works for each algorithm for causal clustering framework.
>
>     -  The problem of causal clustering poses challenges which do not appear in the previous studies, as we cluster on unknown counterfactual functionals that is to be estimated nonparametrically. Surprisingly, a plug-in approach does NOT always works without strong extra conditions. Kim et al. (2024) showed that, in the case of k-means, even the plug-in estimator will fail without the margin condition, which requires local control of the probability around the Voronoi boundaries; without such structural assumptions, the error cannot be bounded.
>     - The main contributions of our work are found in Sections 3 and 4, where we prove that the two appealing off-the-shelf cluster-analysis techniques can be successfully adopted in the framework of causal clustering using simple plug-in methods without requiring extra strong structural assumptions as opposed to k-means, which requires the margin condition (kim et al. 2024). As outlined in the proof, verifying this is more complicated than one may think. (We will give a brief exposition why this could be a theoretically difficult task in the revised text.) Our plug-in approaches could also be readily extended to clustering with generic pseudo-outcomes, even outside the context of causal inference. This versatility may be considered an additional contribution.
>
> 3. **[Presentation]** As mentioned earlier, and as suggested by the reviewer, we will ensure that our revised paper is more self-contained. We believe this will help clarify the novelty of the problem and our key contribution, ensuring that readers do not experience any confusion.
>
> 4. **[Ans to Question]** Thank you for this insightful question. There are two types of error we consider in our work: estimation error of the unknown counterfactual functionals (i.e., the identified regression functions $\{\mu_a\}$) and error regarding to clustering accuracy. I guess your question is related to the latter, or more specifically, whether one may adopt other types of clustering algorithms that can incorporate uncertainty on the hierarchy that is being found. We completely agree that this could lead to fascinating future work, and we will include it in the revised manuscript's discussion section.
>
>
> We hope the above response addresses your concerns. However, if there are any remaining issues, please feel free to let us know through your comments.

---

> ### Comment · Reviewer_geg3 · 2024-08-12
> **ACK of responses**
>
> The reviewer has read the response and thanks tha authors for their time. However given the presentation issues, and the weaknesses raised, the reviewer still thinks the paper should be improved before publications and that it is not ready as is.

---

> ### Author Response · Authors · 2024-08-14
> **Thank you**
>
> Thank you for your input. We believe incorporating your comments on presentation will improve the quality of the paper.

---

### Decision · Program_Chairs · 2024-09-25

**Decision:**

Accept (poster)

**Comment:**

This paper addresses the important problem of treatment effect heterogeneity.  It does not present a new algorithm, but relies broadly on causal k-means clustering, and it presents a new and improved theoretical analysis that contributes insights.  There was a robust discussion with reviewers.  Initially two were positive and two were not, and one of the two who initially were not was swayed to be the most positive of all.  The authors should address the concerns of all reviewers in the revision, and especially should make the changes they described in their replies.  Most notably, they should rewrite the background text about which there was some discussion with reviewers, both to make the paper more self-contained and to not use text that previously appeared elsewhere.  There was also nice discussion with reviewers about possible next steps, and this would be nice to see summarized in a conclusions or future work section.